# STRCMP: Integrating Graph Structural Priors with Language Models for Combinatorial Optimization

**Xijun Li[1,2], Jiexiang Yang[2], Jinghao Wang[2], Bo Peng[1,2], Jianguo Yao[1,2], Haibing Guan[1,2]**

[1]Shanghai Key Laboratory of Scalable Computing and Systems
[2]School of Computer Science, Shanghai Jiao Tong University
{lixijun,jianguo.yao}@sjtu.edu.cn

## Abstract

Combinatorial optimization (CO) problems, central to operation research and theoretical computer science, present significant computational challenges due to their $\mathcal{NP}$-hard nature. While large language models (LLMs) have emerged as promising tools for CO—either by directly generating solutions or synthesizing solver-specific codes—existing approaches often *neglect critical structural priors inherent to CO problems*, leading to suboptimality and iterative inefficiency. Inspired by human experts' success in leveraging CO structures for algorithm design, we propose STRCMP, a novel structure-aware LLM-based algorithm discovery framework that systematically integrates structure priors to enhance solution quality and solving efficiency. Our framework combines a graph neural network (GNN) for extracting structural embeddings from CO instances with an LLM conditioned on these embeddings to identify high-performing algorithms in the form of solver-specific codes. This composite architecture ensures syntactic correctness, preserves problem topology, and aligns with natural language objectives, while an evolutionary refinement process iteratively optimizes generated algorithm. Extensive evaluations across Mixed Integer Linear Programming and Boolean Satisfiability problems, using nine benchmark datasets, demonstrate that our proposed STRCMP outperforms five strong neural and LLM-based methods by a large margin, in terms of both solution optimality and computational efficiency. The code is publicly available in the repository: https://github.com/Y-Palver/L2O-STRCMP.

## 1 Introduction

Making complex plans subject to multiple constraints is a time- and labor-intensive process, but is critical in many aspects of our lives such as scheduling [1], logistics [2], and robotics [3]. These problems are frequently modeled as combinatorial optimization (CO) problem, a cornerstone of operation research and theoretical computer science, where the objective is to identify optimal solutions within discrete, highly constrained search spaces. However, the inherent $\mathcal{NP}$-hardness of many CO problems renders obtaining exact solutions computationally intractable, posing significant challenges in solving them efficiently and accurately. Consequently, substantial research efforts have endeavored to developing algorithms that balance solution quality with computational cost, typically requiring specialized domain expertise and rigorous analytical modeling of problem structure [4–6].

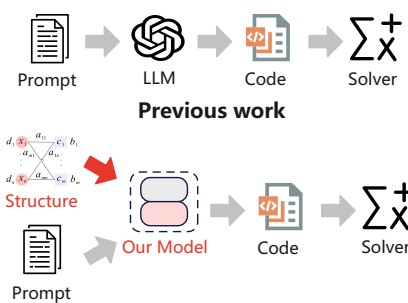

Figure 1: Intuitive comparison between prior work and the proposed framework.

39th Conference on Neural Information Processing Systems (NeurIPS 2025).

Machine learning (ML) techniques, particularly deep learning and reinforcement learning, have demonstrated significant potential in addressing CO problems [7]. As many CO problems arising from similar application domains exhibit inherent structural patterns, ML methods can leverage these patterns to reduce computational complexity in traditional CO approaches through data-driven strategies. The integration of ML and CO frameworks can be broadly categorized into three paradigms[7]: (1) End-to-end neural solvers [8–10], which train ML models to directly output CO solutions; (2) ML-augmented algorithm configuration [11–13], leveraging ML models to predict optimal algorithmic configurations for specific CO algorithm to solve associated problems; and (3) Hybrid methods [14–16], embedding ML models as critical decision modules within traditional CO solvers. However, real-world adoption of above approaches remains limited due to reliance on training distributional alignment for generalization and inherently insufficient interpretability [17].

Building on the explosive advancements in large language models (LLMs) over recent years, combinatorial optimization tasks have shown potential for delegation to LLMs, particularly owing to their improved accountability and interpretability compared to conventional ML approaches. Capitalizing on their extensive world-knowledge priors and advanced code generation capabilities, numerous LLM-based approaches for CO problems have emerged. These methods bifurcate into two classes: (1) direct utilization of LLMs to solve CO problems [18, 19], and (2) employing LLMs to discover high-quality algorithm (in the form of solver-specific codes) for traditional solvers [20–22]. For example, Yang et al. [18] present Optimization by PROmpting (OPRO), which harnesses LLMs as black-box optimizers through iterative solution refinement via natural-language problem specifications and optimization trajectories. In contrast, Romera-Paredes et al. [20] proposes FunSearch that integrates a pre-trained LLM into an evolutionary algorithm to incrementally generate solver code, achieving state-of-the-art results on the cap set problem and discovering novel heuristics for online bin packing. Additionally, Huang et al. [19] demonstrates that multi-modal fusion of textual and visual prompts enhances LLM-driven optimization performance, as evidenced in capacitated vehicle routing problems.

Despite these advances, current LLM-based approaches for solving CO problems remain in their infancy. As illustrated in Figure 1, they primarily rely on textual (or occasionally visual) prompting mechanisms to interface with LLMs, *failing to effectively exploit the inherent topological structures of CO problems*. Historically, human experts have successfully leveraged structural priors of CO problems to design sophisticated and efficient algorithms [4–6]. Furthermore, existing LLM-based methods [20, 21, 18] often require multiple iterations to produce high-quality solutions or algorithm implementations, primarily due to their lack of integration with CO-specific structural priors. As highlighted in Yao's insightful analysis [23], pre-training language model establishes strong priors for conversational tasks but weaker priors for structured domains like computer controls or video games — *challenges exacerbated in solving CO problems, as these domains differ significantly from Internet text distributions.* Given these challenges, developing a generative model that effectively integrates structural priors for CO problem solving becomes particularly compelling. As shown in Figure 1, the distinct insight lies in constructing a generative model that generates solver-specific code while respecting the intrinsic topological structure of CO problems. Theoretically, this approach aligns with the rising research directions of multi-modal generative model. Practically, such integration offers substantial benefits, dramatically reducing inference iterations while enhancing solution quality in solving CO problem.

To tackle this challenge, we present STRCMP [1], a novel structure-prior-aware LLM-based algorithm discovery framework for solving CO problems. To the best of our knowledge, STRCMP is the first framework to explicitly integrate structural priors of CO problems into LLM-driven algorithm discovery, jointly improving solution quality and computational efficiency. Our framework first constructs a composite architecture combining a graph neural network (GNN) and an LLM. The GNN extracts structural embeddings from input CO instances, which serves as inductive biases for subsequent algorithm discovery via code generation. The LLM then produces solver-specific code conditioned on these structural priors, ensuring adherence to solver syntax, preservation of CO problems' topological characteristics, and alignment with natural language optimization objectives. This composite model is further embedded within an evolutionary algorithm-based refinement process to iteratively enhance solution quality. Besides, we provide the theoretical analysis why fusing the structure prior into LLM can benefit solving CO problem from the perspective of information theory.

---

[1] STRCMP stands for STRucture-aware CoMPposite model that can discover and generate algorithm implementation for CO problems considering intrinsic topological structure of the problem.

Extensive experiments across two representative CO domains (Mixed Integer Linear Programming and Boolean Satisfiability), spanning nine benchmark datasets, demonstrate that `STRCMP` outperforms five well-recognized baselines by a large margin—including neural combinatorial optimization methods and LLM-based approaches—in both solution optimality and computational efficiency.

## 2 Related Work

**Neural Combinatorial Optimization.** Extensive research [14, 24, 25, 15, 16] has explored machine learning integration with combinatorial optimization for improved computational approaches to $\mathcal{NP}$-hard problems, forming Neural Combinatorial Optimization (NCO) methods that either approximate expert heuristics or learn policies via reinforcement learning. Following Bengio et al. [7]'s taxonomy, NCO encompasses: (1) End-to-end neural solvers [8–10], (2) ML-augmented algorithm configuration [11–13], and (3) Hybrid methods integrating learned components into classical frameworks [14–16]. Pointer Networks [25] represent seminal work, employing attention mechanisms [8] as dynamic pointers for CO problems, eliminating dependence on fixed output dictionaries in conventional seq2seq models [26]. The framework achieves competitive performance on convex hulls, Delaunay triangulation, and small TSP instances while generalizing to unseen sizes. In hybrid approaches, Gasse et al. [14] introduced GCNs trained via imitation learning to replace strong branching heuristics [27] in branch-and-bound algorithms for MIP solvers. While avoiding manual feature engineering through graph representations, such approaches face scalability challenges under extreme problem sizes, a key limitation of neural solver architectures. Moreover, NCO's reliance on opaque machine learning models yields policies with insufficient interpretability—problematic for high-stakes decision-making systems demanding traceable logic [17]. Our approach addresses these gaps by furnishing optimality certificates or bounded sub-optimality guarantees via CO solver integration. By procedurally generating solver-executable code through iterative refinement, our framework ensures superior interpretability relative to black-box NCO policy learning.

**Language Models for Combinatorial Optimization Problem.** Recent advances in large language models (LLMs) have spurred explorations of their applications to combinatorial optimization [20, 21, 28–30]. Current approaches fall into two categories: (1) direct solution generation via LLM inference [18, 19], and (2) automated generation of solver-compatible formalizations [20–22, 30]. In the first paradigm, Yang et al. [18] propose Optimization by PROmpting (OPRO), leveraging LLMs as black-box optimizers through iterative refinement of natural-language problem descriptions. While achieving comparable performance to heuristics on small TSP instances, OPRO scales poorly beyond 50 nodes due to context limitations and solution-space complexity. For the second paradigm, Romera-Paredes et al. [20] introduce FunSearch, combining evolutionary methods with LLM-guided program synthesis to obtain breakthroughs on the cap set problem [31] and generate novel bin packing heuristics [32]. Similarly, [21] develop Evolution of Heuristics (EoH), integrating evolutionary algorithms with LLMs to co-optimize natural language "thoughts" and code implementations through iterative prompting, outperforming existing methods like FunSearch in query efficiency [20]. Existing approaches predominantly employ natural language (or occasionally visual [19]) prompts combined with evolutionary frameworks for iterative code generation to solve CO problems. However, *these methods universally neglect the topological structural characteristics intrinsic to combinatorial optimization problems*, which are systematically exploited by human experts during algorithm design [4–6]. Furthermore, evolutionary framework-based code generation with LLMs often necessitates multiple iterations for convergence while incurring significant computational overhead from repeated solver invocations during evaluation. To overcome these limitations, we propose a composite model that effectively incorporates structural priors of CO problems during algorithmic discovery. By developing composite architectures tailored to CO problem structures, our method substantially improves both solution quality and computational efficiency compared to existing LLM-based solving approaches.

## 3 Problem Statement

**Combinatorial Optimization.** Following [33], we formulate a general combinatorial optimization problem $Q$ (*e.g.*, Boolean Satisfiability (SAT) and Mixed Integer Linear Programming (MILP)) as a constrained discrete optimization problem. For an instance $q$ of $Q$, the formulation becomes:

$$\min_{x \in S(q)} c(x;q) + f(x;q), \tag{1}$$

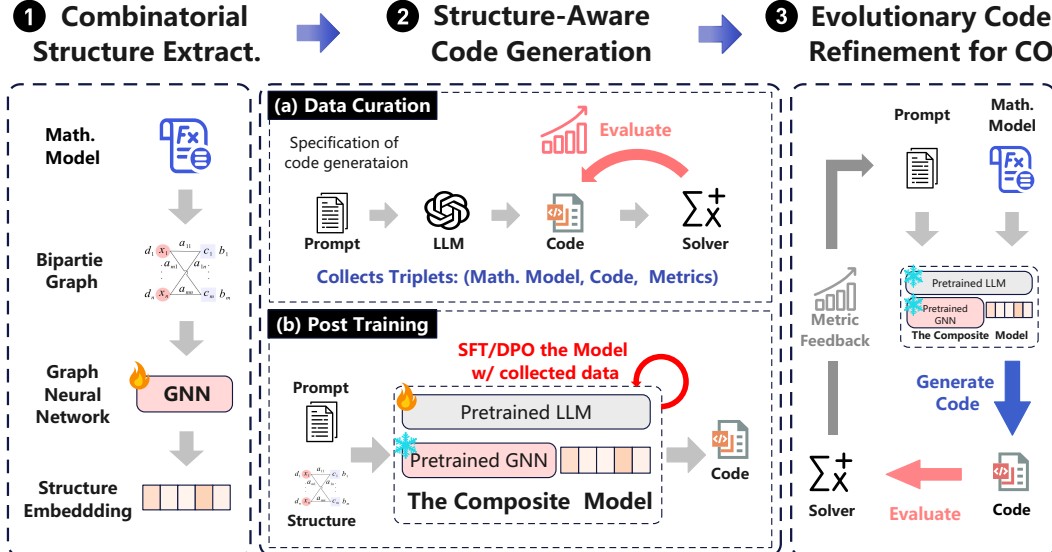

Figure 2: Overview of the proposed STRCMP. ❶ **Combinatorial Structure Extraction.** We utilize a graph neural network (GNN) to encode the topological structure of combinatorial optimization problems into latent embeddings, capturing problem-specific structural invariants. ❷ **Structure-Aware Code Generation:** (a) Data Curation: For a given CO problem's mathematical model, target solver and the specific prompt, an LLM generates candidate algorithm in the form of code snippets, which are automatically validated via the solver execution to generate performance metrics. This yields a curated dataset comprising (mathematical model, code snippet, metric) triplets. (b) Post Training: A composite architecture integrating a frozen pretrained GNN with an LLM is fine-tuned on this dataset, ensuring code generation respects both the CO's intrinsic topology and solver-specific syntax. ❸ **Evolutionary Code Refinement.** Drawing on [20], the composite model is embedded within an evolutionary framework to evolve the algorithm discovery process through performance-driven iterations.

where $x$ denotes decision variables subject to solution space $S(q)$; $c(x;q)$ represents the objective function to minimize, and $f(x;q)$ corresponds to constraint penalties (zero when all constraints are satisfied). In SAT problems, the objective function $c(x;q)$ is absent since only constraint satisfaction is required. Conversely, MILP problems feature both objective and constraint functions.

**Generation Task.** Given a combinatorial optimization problem $Q$, our goal is to find an algorithm $A$ that can consistently solve $Q$ well:

$$\min_{A \sim \mathcal{A}} \mathbb{E}_{q \sim Q, x \sim A(q)}[c(x;q) + f(x;q)] \tag{2}$$

where $\mathcal{A}$ denotes the algorithm search space. This work focuses on symbolic search spaces where $\mathcal{A}$ comprises algorithms representable as code snippets executable within a combinatorial optimization solver. We resort to a generative model to synthesize the target algorithm through code generation.

## 4 Proposed Solution

The proposed framework is illustrated in Figure 2. We first describe the technical interplay among components within the proposed framework, highlighting design motivations and synergistic interactions. Subsequently, we provide a theoretical analysis demonstrating that integrating structural priors into LLMs enhances performance in solving combinatorial optimization problems.

### 4.1 Methodology

❶ **Combinatorial Structure Extraction.** Combinatorial optimization problem is equivalently converted to a bipartite graph. Thus, the intrinsic structure of the combinatorial optimization problem can be easily captured by a graph neural network, as has been done in many previous work [16, 24, 14].

As shown in the leftmost part of Figure 2, we continue to use this technique to extract structure embedding of given CO problem, benefiting the following algorithm discovery process.

Specifically, a bipartite graph $G = (C, E, V)$ is constructed for the CO problem instance $q$, where $C$ corresponds to the constraints of $q$; $V$ denotes the variables of $q$; and an edge $e_{ij} \in E$ between a constraint node $i$ and a variable node $j$ if they have a connection. A graph neural network $\theta_G$ takes as input the bipartite graph to generate the structural embedding $h_q \in \mathbb{R}^d$ of the problem instance $q$. We simply train $\theta_G$ via a classification task, where a group of CO problems and their class label are given. The problems falling into the same class means that they originates from the same scenario/domain, such as traveling salesman problems [34], production planning [35], vehicle routing problem [36], *etc*. The data representation, training, implementation details are left to Appendix B.1. In this way, GNNs can capture structure priors of CO problem (such as symmetry, sparsity, and degeneracy), which are critical for solver decisions. The extracted embeddings transfers structural insights to downstream code generation task, facilitating generalization across problem classes.

❷ **Structure-Aware Code Generation.** In this step, we construct a composite generative model to achieve structure-aware algorithm discovery for CO problem. The composite model is essentially a concatenation of a GNN and an LLM. GNN is obtained via training method mentioned in Step ❶ and frozen in the composite model, which provides the structure embedding of given CO problem to LLM. The LLM takes as input the natural language description of the problem and of the code generation task. Then the LLM generates candidate algorithm in the form of code snippet conditioned on the natural language description and structure embedding for a specific solver, as shown in the middle part of Figure 2. The code generation procedure can be formulated below:

$$P(w_1, w_2, ..., w_T) = \prod_{t=1}^{T} P_{\theta_L}(w_t | w_{<t}; h_q, NL), \tag{3}$$

where $\{w_i | i = 1, , , T\}$ is the code snippet predicted by the composite model; $h_q$ is the structure embedding of given CO problem instance $q$; $NL$ is the natural language description of $q$ and of code generation task; $\theta_L$ is the parameter of LLM.

As anticipated, the composite model exhibits suboptimal performance on code generation tasks for CO problems due to architectural incompatibilities that disrupt the native token prediction mechanism of the underlying LLM. To address this architectural mismatch, we implement a two-phase training protocol consisting of: (a) **data curation** through systematic problem sampling, and (b) parameter optimization via **post-training**. This phased approach enables effective adaptation of the composite architecture while preserving the structural integrity of the original CO problem representations.

**(a) Data Curation.** To facilitate the post-training phase, we aim to collect four key categories of data: 1) mathematical formulations of CO problems; 2) natural language specifications detailing problem requirements alongside corresponding code generation objectives for targeted CO solvers; 3) executable code implementations derived from these specifications; and 4) quantitative performance metrics evaluating solution quality and computational efficiency of the implemented code.

**(b) Post-Training.** In this phase, we only train the parameter $\theta_L$ of the composite model, which means that structural embeddings remain consistent during post-training. We conduct Supervised Fine-Tuning (SFT) on $\theta_L$ using the curated dataset, followed by Direct Preference Optimization (DPO) [37] initialized from the SFT checkpoint to derive the final composite model. All details, including prompt template, data curation and post-training, are presented in Appendix B.2.

❸ **Evolutionary Code Refinement for Combinatorial Optimization Problem.** Prior work [30, 20, 29] leverages LLMs' code generation and algorithm design capabilities within Evolutionary Algorithms (EAs) to address combinatorial optimization problems through iterative feedback frameworks. The iterative nature of evolutionary algorithms introduces significant computational overhead as the primary limitation of prior approaches [30, 20, 29] for identifying optimal code snippets or algorithms to solve combinatorial optimization problems. Convergence typically requires numerous iterations, exacerbated by the prolonged execution times required for solver-based evaluation of generated code or algorithm candidates.

Our evolutionary code optimization framework for combinatorial optimization adopts the core principles of prior work[30, 20, 29], with the significant distinction lying in the composite model learned in Step ❷ (see Figure 2 right panel). Our framework jointly processes both textual problem descriptions with algorithm discovery objectives and formal mathematical model of CO problems.

The composite model generates solver-specific, structure-aware code through standard EA operators (selection, crossover, mutation), producing higher-quality algorithm implementation within fewer iterations – a capability empirically validated in our experiments. By integrating this model into EA-based algorithm discovery frameworks, we achieve superior optimization performance while maintaining compatibility with existing evolutionary algorithm design paradigms, suggesting broader applicability across LLM-driven optimization methodologies.

## 4.2 Theoretical Analysis

In the following, we give the theoretical analysis why fusing the structure prior into the generative model helps algorithm discovery for solving CO problem on the basis of information theory [38] and multi-modal co-learning [39]. Specifically, we prove that a generative model with an additional prior will not lower the upper bound of its performance on the downstream task. Note that we leave all proofs into the Appendix A due to the space limitation.

**Definition 1** (Upper Bound of Model Performance). *The performance upper bound of generative model, denoted by $\sup(\mathcal{P})$, is utilized to measure the maximum expected performance of a generative model for a downstream task $p(\mathbf{w}|\mathbf{c})$, where $\mathbf{w}$ is the generated content conditional on the different kinds of prior $\mathbf{c}$. Formally, $\sup(\mathcal{P})$ associated with kinds of prior $\mathcal{C}$ can be expressed as*

$$\sup(\mathcal{P}_{\mathcal{C}}) = \sum_{\mathbf{C} \in \mathcal{C}} \sum_{c \in \mathbf{C}} p(\mathbf{c}) \max_{\mathbf{w}} p(\mathbf{w}|\mathbf{c}) \Phi(\mathbf{w}), \tag{4}$$

*where $\Phi$ is the performance evaluator for given generated content $\mathbf{w}$; $\mathcal{C}$ is the complete set of different priors and $\mathbf{C}$ is a type of prior belonging to the prior set $\mathcal{C}$.*

**Theorem 1.** *Given a prior dataset $\mathcal{D}$ whose data samples comprises $M$ types of prior $\mathcal{C} = \{\mathbf{C}_1, ..., \mathbf{C}_M\}$, the distinct priors follow respective true distribution $p(\mathbf{C}_i|\mathbf{w})$. Let $\mathbf{c}_i$ be a sample drawn from the distribution, i.e., $\mathbf{c}_i \sim p(\mathbf{C}_i|\mathbf{w})$. A generative model with an additional type of prior will not lower the upper bound of model performance $\sup(\mathcal{P})$.*

**Remark 1.** *Theorem 1 establishes that introducing additional priors cannot reduce a generative model's performance upper bound, while decreasing its entropy. Consequently, integrating combinatorial optimization structural priors into LLMs does not degrade their performance in generating code for solving CO problems.*

**Definition 2** (Performance-Enhancing Prior). *Assume a type of prior $\mathbf{C}$ can boost the performance of a generative model compared to the one without the prior, then $\mathbf{C}$ is the performance-enhancing prior to the generative model. It can be formally expressed as*

$$\exists \mathbf{w}' \neq \mathbf{w}^*, p(\mathbf{w}'|\mathbf{c})\Phi(\mathbf{w}')_{\mathbf{c} \in \mathcal{C}} > p(\mathbf{w}^*|\mathbf{c})\Phi(\mathbf{w}^*)_{\mathbf{c} \in \mathcal{C} \setminus \{\mathbf{C}\}}, \tag{5}$$

*where $\mathbf{w}^* = \arg\max_{\mathbf{w}} p(\mathbf{w}|\mathbf{c})\Phi(\mathbf{w})_{\mathbf{c} \in \mathcal{C} \setminus \{\mathbf{C}\}}$.*

**Theorem 2.** *If prior $\mathbf{C}_{pe}$ is a performance-enhancing prior, a generative model neglecting prior $\mathbf{C}_{pe}$ will decrease the upper bound of model performance. It can be expressed as*

$$\sup(\mathcal{P}_{\mathcal{C} \setminus \{\mathbf{C}_{pe}\}}) < \sup(\mathcal{P}_{\mathcal{C}}) \tag{6}$$

**Remark 2.** *Theorem 2 demonstrates that generative models equipped with performance-enhancing priors achieve superior performance relative to their prior-free counterparts. Specifically, the structural prior serves as such performance-enhancing prior for LLMs in code generation tasks targeting CO problems, which is empirically validated through our comprehensive experiments.*

## 5 Experimental Evaluation

To empirically evaluate the efficacy of our proposed STRCMP in solving combinatorial optimization problems, we conduct extensive experiments across two fundamental CO problem classes: mixed-integer linear programming and Boolean satisfiability, evaluated on over nine benchmark datasets. We benchmark our approach against five well-established baselines encompassing both neural optimization methods and contemporary LLM-based approaches. We aim to answer the following research questions:

**Research Questions**

**RQ1:** Does the proposed `STRCMP` identify superior algorithmic implementations compared to existing algorithm discovery approaches?
**RQ2:** Does the proposed composite model effectively reduce computational overhead in existing algorithm discovery frameworks?
**RQ3:** Does the structural prior benefit the generative model in solving combinatorial optimization problems?

## 5.1 Settings

**Baselines.** We evaluate our framework against two categories of baselines: neural combinatorial optimization methods and LLM-based evolutionary code optimization frameworks, covering mixed-integer linear programming (MILP) and Boolean satisfiability (SAT). More details of baselines and used backend solvers can be found in Appendix C.

- *Neural Combinatorial Optimization*: For MILP, we compare with the seminal work **L2B** [14], which employs graph convolutional networks for variable selection to replace the strong branching policy, and **HEM** [15, 16], a hierarchical sequence model for cut selection in branch-and-bound solvers. For SAT, we include **NeuroSAT** [40], a message-passing neural network trained with single-bit supervision for SAT solving.

- *Evolutionary Code Optimization*: While numerous LLM-based evolutionary code optimization approaches exist for combinatorial optimization, we specifically compare with two methods specialized for SAT and MILP: **AutoSAT** [29] and **LLM4Solver** [30]. **AutoSAT** leverages LLMs to automate heuristic optimization in SAT solvers, minimizing manual intervention. **LLM4Solver** integrates LLMs with multi-objective evolutionary algorithms to automatically design effective diving heuristics for MILP solvers[2].

**Dataset.** We perform the empirical evaluation over ten widely-used benchmark dataset for SAT and MILP respectively. More details and statistics of the used datasets can be found in Appendix D.

- *Mixed-integer linear programming (MILP)*: The datasets for MILP solver evaluation contain three difficulty tiers [15, 16, 30]: (1) **Easy** features synthetic benchmarks (Set Covering [41], Maximum Independent Set [42], Multiple Knapsack [43]) generated using protocols from [44, 45]; (2) **Medium** includes MIK [46] and CORLAT [47]; (3) **Hard** contains the Google-inspired Load Balancing problem and the industrial-scale Anonymous problem [48].

- *Boolean satisfiability (SAT)*: The dataset comprises two sources: SAT Competition problems [49] and automatically generated instances via Picat [29]. **SAT Competition data** includes Profitable-Robust-Product (PRP) and Chromatic-Number-of-the-Plane (CNP) problems, while **Picat-generated data** contains CoinsGrid and Zamkeller instances. We adhere to the generation protocol established in [29].

**Metrics.** To evaluate the effectiveness of the proposed framework, we analyze solving time to optimality or solution quality attainable within a fixed time budget. For solving efficiency evaluation, we measure the number of iterations or training steps required for convergence. Specifically, for MILP domain, critical metrics include **solving time** and **primal-dual (PD) integral** which are widely used in benchmarking the MILP solvers [15, 16, 14]. For SAT domain, key metrics encompass **solving time, PAR-2, and number of timeout** frequently measured in evaluating SAT solvers [29, 40]. Metrics such as solving time, PD integral, number of timeout, and PAR-2 are minimized through optimization. More details of the used metrics are presented in Appendix E.

## 5.2 Implementation

**Model Architecture.** The proposed composite model comprises two components: a GNN and a structure-prior-aware LLM. We implement the GNN using graph convolution operators from `torch_geometric`, structured with three sequential convolutional layers terminated by global mean

---

[2]For fair comparison, we slightly adjusted **LLM4Solver**'s optimization focus from diving heuristics to cut selection for MILP solver.

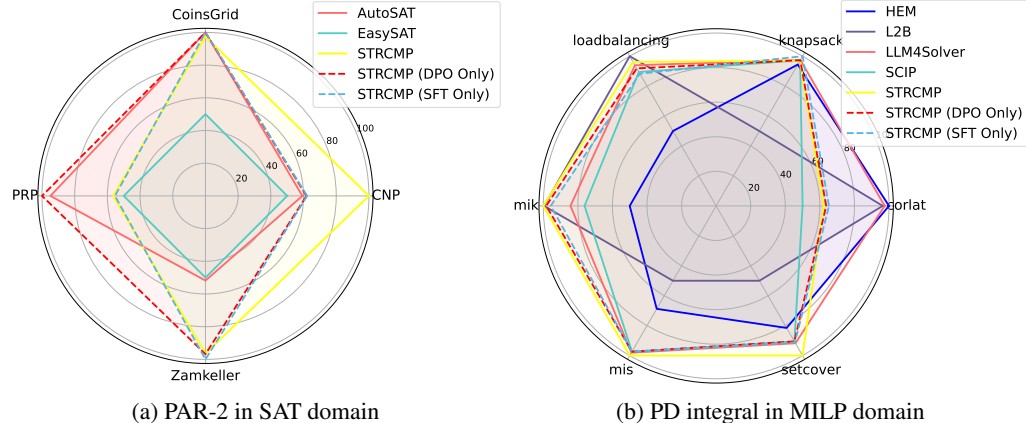

(a) PAR-2 in SAT domain       (b) PD integral in MILP domain

Figure 3: Optimization performance over different CO domains. Aligned with [29], we normalize each benchmark's indicator values using $1 - \frac{value-min}{2\times(max-min)}$, where $value$ represents the method's measured indicator, with $min$ and $max$ denoting respectively the minimum and maximum values observed during evaluation. A larger shaded area corresponds to superior performance.

pooling. We adapt the LLM component based on `Qwen2.5-Coder-7B-Instructor` model through modifying its architecture, forward propagation dynamics and inference paradigm. Comprehensive implementation details including hyperparameter configurations, adaptations, and software dependencies are provided in Appendix B.1 and B.2.

**Training, Inference and Used Hardware.** We first train domain-specific GNNs for SAT and MILP problems using aforementioned benchmark dataset. The GNNs are trained for three epochs on SAT instances and four epochs on MILP instances, followed by conducting post-training of the adapted LLM with a specialized corpus. All post-training data is collected via queries to `Qwen2.5-Coder-7B-Instructor`, aligned with the model used in the post training. The adapted LLM undergoes three epochs of post training per domain. Both GNN and LLM selecting optimal checkpoints based on validation prediction loss. We then integrate the composite model (*i.e.* the trained GNN and adapted LLM) into an EA-based framework to discover solver-specific algorithm configurations optimized for training instances. Finally, we evaluate the performance of identified algorithms on held-out test sets. All experiments are conducted on hardware platform with dual AMD EPYC 9534 64-core processors @ 2.45GHz and two NVIDIA H800 80GB GPUs connected via PCIe. More training specifics and data curation are detailed in Appendix B.1 and B.2 respectively.

### 5.3 Results

**Optimization Performance (Answer to RQ1).** The comparative optimization results are presented in Figure 3 and Table 1 ("CGD" and "ZAM" denotes Coins-Grid and Zamkeller dataset respectively) with respect to primary objective metrics (*i.e.* PAR-2 and Number of Timeout for SAT and PD integral for MILP domains). Comprehensive results for all evaluation metrics are provided in Appendix G.1). As evidenced in Figure 3 and Table 1, our STRCMP with various post-training variants consistently matches or exceeds all baseline performance across both SAT and MILP domains. Specifically for SAT, STRCMP demonstrates universal superiority over its closest counterpart AutoSAT, particularly achieving significant reductions in terms of timeout (77.8% on Zamkeller: 18→4; 66.7% on PRP: 9→3) and solving time (on PRP: 22967 seconds → 21146 seconds; on

Table 1: Optimization performance result w.r.t. Number of Timeout between different methods over SAT domain.

| Compared Methods | Number of Timeout ($\downarrow$) | | | |
|---|---|---|---|---|
| | CNP | CGD | PRP | ZAM |
| AutoSAT [29] | 32 | 16 | 9 | 18 |
| EasySAT | 32 | 25 | 50 | 18 |
| NeuroSAT [40] | 44 | 18 | 46 | 29 |
| STRCMP | **31** | 17 | 44 | 5 |
| STRCMP (DPO Only) | 32 | **16** | **3** | 5 |
| STRCMP (SFT Only) | 33 | **16** | 45 | **4** |
| STRCMP w/o GNN | 32 | **16** | 44 | 14 |

Zamkeller: 20772 seconds → 6929 seconds). On MILP benchmarks, STRCMP maintains strong performance parity with NCO methods L2B/HEM and its direct competitor LLM4Solver. These

empirical findings conclusively address RQ1: *STRCMP successfully identifies superior algorithmic configurations compared to existing strong baselines.*

**Efficiency Improvement (Answer to RQ2).** Next, we assess the efficiency of our proposed STRCMP framework in discovering high-performance algorithms via iterative search for combinatorial optimization problems. The representative convergence comparison for SAT domain are presented in Figures 4. More evaluations over other SAT datasets and MILP domain across additional metrics are provided in Appendix G.2. Our experiments reveal that STRCMP achieves convergence significantly faster than existing evolutionary-based algorithm discovery frameworks. Furthermore, the framework

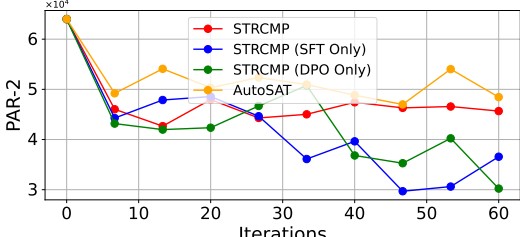

Figure 4: Convergence comparison (w.r.t. PAR-2) between evolutionary-based algorithm discovery frameworks on Zamkeller dataset of SAT domain.

attains markedly higher-quality convergence points compared to baseline methods AutoSAT and LLM4Solver. Notably, STRCMP achieves stable convergence while AutoSAT exhibits persistent oscillations even after convergence. These findings validate that *our proposed STRCMP substantially reduces computational overhead in evolutionary-based algorithm discovery frameworks.* **Ablation Studies (Answer to RQ3).** To address RQ3, we conduct comprehensive ablation studies on our composite model by systematically deactivating individual components and evaluating their impact on optimization performance and computational efficiency. Representative results are shown in Figure 5, with full ablation studies detailed in Appendix G.3. Analysis of Figure 5 reveals that *the structural prior provides measurable benefits for code generation in combinatorial optimization tasks: the STRCMP w/o GNN variant (lacking structural guidance) exhibits consistently inferior optimization performance compared to counterparts incorporating the prior.* Furthermore, this variant demonstrates increased solution variability during algorithmic search iterations, potentially resulting in higher computational costs. Counterintuitively, the full STRCMP model (with complete post-training) does not uniformly outperform its ablations STRCMP (SFT Only) and STRCMP (DPO Only) across all benchmarks, suggesting underlying conflicts within the post-training data distribution. We plan to investigate this phenomenon in subsequent research.

## 5.4 Further Analysis

**Two-phase Training v.s. Joint Training?** Besides the implementation of two-phase training strategy, we also design and implement an end-to-end joint training method for our proposed model. This new method directly aligns the learning objectives of the graph and text modalities with the same optimization goal: improving the performance of the generated code. Specifically, we do not discriminate between the weights of the GNN and the LLM, meaning they share the same loss function, and the gradient is backpropagated from the parameters of the LLM to those of the GNN. We implemented the end-to-end joint training method described above and tested the resulting models on two SAT datasets (i.e., Zamkeller and PRP). All the experimental settings keep the same except

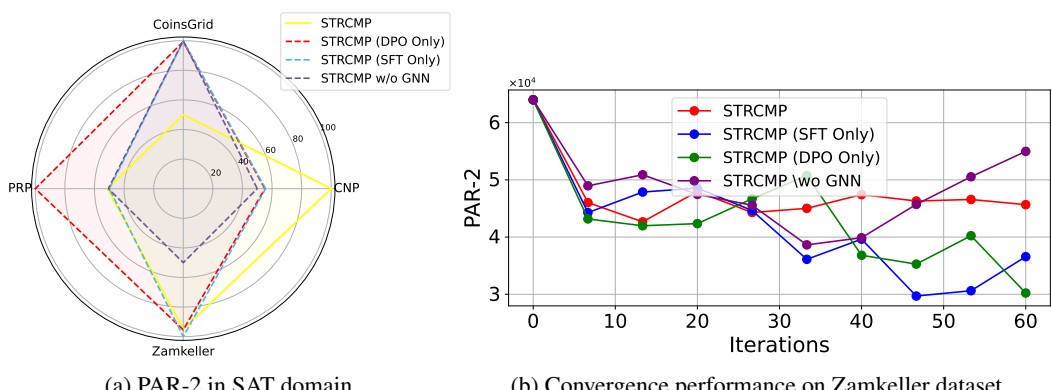

(a) PAR-2 in SAT domain       (b) Convergence performance on Zamkeller dataset

Figure 5: Ablation studies in terms of optimization performance and convergence rate.

for the training method. The detailed results of this new training and testing are presented show in Table 7 and Table 8 in Appendix I. We can conclude the followings from these results: 1) The end-to-end joint training is extremely unstable compared to our proposed two-stage training method. Compared with the joint training, the loss curve of our two-stage training method is stable and close to zero at convergence. The GNN and LLM are trained together, which makes it difficult for the models to learn optimal parameters simultaneously; 2) The model exhibits poor performance when tested on solving SAT instances. Unsurprisingly, due to its very unstable training and relatively high loss at convergence, the performance of the jointly trained model is much lower than our two-stage model on the two SAT datasets.

**Insight of Training Strategy.** To provide a clearer and more comprehensive answer, we offer a detailed analysis focusing on dataset complexity and the iterative optimization behavior of our proposed method, STRCMP , and its variants, shown in Appendix J. Through the above thorough analysis and additional experiments, we can summarize our insights on selecting a training strategy: 1) For "easy" datasets: The SFT-only approach is often sufficient and efficient. On these problems, high-quality training data (i.e., optimal or near-optimal solutions) can be generated at a low cost, allowing SFT to quickly learn an effective policy. 2) For "hard" datasets: We strongly recommend a DPO-based approach (DPO-only or SFT+DPO). For these problems, obtaining optimal solutions for SFT is prohibitively expensive or impossible. However, generating preference pairs by comparing the performance of different candidate solutions is still feasible and relatively cheap. The superior generalization of DPO-trained models makes them better suited to navigating the vast and complex search spaces of these challenging instances. 3) For medium complexity datasets, the choice is less definitive. As the results show, the interplay between SFT and DPO is intricate. Our full STRCMP model (SFT+DPO) often acts as a robust default, leveraging SFT for a strong initialization and DPO for refinement and generalization.

**Dependence on the Underlying LLM.** To investigate how much does the performance of STRCMP depend on the underlying LLM, we focused our evaluation on the Llama2 and Qwen2 families of models, selecting representatives of varying sizes. We performed training and testing on two datasets from the MILP domain and two from the SAT domain. The full result and analysis are presented in Appendix K. It is important to note that the entire Llama2 family of models failed on our code generation task, either by exceeding the context limitation of 4096 tokens or by being unable to generate syntactically correct, executable code. Similarly, the smaller Qwen2 models (0.5B and 1.5B) were also unable to produce viable code for the solvers. This underscores the complexity of the task, which requires a highly capable code-generating LLM. From the results in Table 18 and Table 19, we can draw the following conclusions: 1) The structural embedding from the GNN consistently and significantly contributes to the final performance; 2) STRCMP 's performance is robust, provided a sufficiently capable LLM backbone is used, and 3) STRCMP 's performance is sensitive to the size of the LLM backbone, but this effect is problem-dependent.

## 6 Conclusion

We propose STRCMP, a novel structure-aware LLM-based algorithm discovery framework for combinatorial optimization problems. To our knowledge, this represents the first methodology that explicitly incorporates structural priors of combinatorial optimization problems into LLM-driven algorithm discovery, simultaneously enhancing solution quality and computational efficiency. We validate the effectiveness of STRCMP through theoretical analysis and extensive empirical evaluations across SAT and MILP problems, demonstrating consistent improvements in both solving efficiency and solution optimality. Our framework offers a principled foundation for future research integrating additional modal priors into LLMs for solving combinatorial optimization problems.

## Acknowledgements

We thank all the reviewers and chairs for their reviews and valuable feedback. This work was funded by the National Key Research & Development Program of China (No. 2022YFB4500103), NSFC (No. 62506227, 62032008), STCSM (No. 25ZR1402224, 23511100100), Startup Fund for Young Faculty at SJTU (SFYF at SJTU, No. 25X010502616) and Aviation Key Laboratory of Science and Technology on Aerospace Vehicle. The corresponding authors are Bo Peng and Jianguo Yao.

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

# A Proofs

**Theorem 1.** *Given a prior dataset $\mathcal{D}$ whose data samples comprises $M$ types of prior $\mathcal{C} = \{\mathbf{C}_1, ..., \mathbf{C}_M\}$, the distinct priors follow respective true distribution $p(\mathbf{C}_i|\mathbf{w})$. Let $\mathbf{c}_i$ be a sample drawn from the distribution, i.e., $\mathbf{c}_i \sim p(\mathbf{C}_i|\mathbf{w})$. A generative model with an additional type of prior will not lower the upper bound of model performance $\sup(\mathcal{P})$.*

*Proof.* Assume that there is a generative model, which is associated with a set of prior $\mathcal{C}_K = \{\mathbf{C}_1, ..., \mathbf{C}_K\}$. Its upper bound of model performance is denoted by $\sup(\mathcal{P}_{\mathcal{C}_K})$. According to Definition 1, $\sup(\mathcal{P}_{\mathcal{C}_K})$ can be expanded as

$$
\begin{aligned}
\sup(\mathcal{P}_{\mathcal{C}_K}) &= \sum_{\mathbf{C} \in \mathcal{C}_K} \sum_{\mathbf{c} \in \mathbf{C}} p(\mathbf{c}) \max_{\mathbf{w}} p(\mathbf{w}|\mathbf{c}) \Phi(\mathbf{w}) \\
&= \sum_{\mathbf{C} \in \mathcal{C}_K} \sum_{\mathbf{c} \in \mathbf{C}} p(\mathbf{c}) \max_{\mathbf{w}} \sum_{\mathbf{c}_i \in \mathbf{C}_i} p(\mathbf{c}_i|\mathbf{c}) p(\mathbf{w}|\mathbf{c}, \mathbf{c}_i) \Phi(\mathbf{w}) \\
&\leq \sum_{\mathbf{C} \in \mathcal{C}_K} \sum_{\mathbf{c} \in \mathbf{C}} \sum_{\mathbf{c}_i \in \mathbf{C}_i} p(\mathbf{c}) p(\mathbf{c}_i|\mathbf{c}) \max_{\mathbf{w}} p(\mathbf{w}|\mathbf{c}, \mathbf{c}_i) \Phi(\mathbf{w}) \\
&= \sum_{\mathbf{C} \in \mathcal{C}_K} \sum_{\mathbf{c} \in \mathbf{C}} \sum_{\mathbf{c}_i \in \mathbf{C}_i} p(\mathbf{c}_i, \mathbf{c}) \max_{\mathbf{w}} p(\mathbf{w}|\mathbf{c}, \mathbf{c}_i) \Phi(\mathbf{w}) \\
&= \sum_{\mathbf{C} \in \mathcal{C}_K \bigcup \{\mathbf{C}_i\}} \sum_{\mathbf{c} \in \mathbf{C}} p(\mathbf{c}_i, \mathbf{c}) \max_{\mathbf{w}} p(\mathbf{w}|\mathbf{c}, \mathbf{c}_i) \Phi(\mathbf{w}) \\
&= \sup(\mathcal{P}_{\mathcal{C}_K \bigcup \{\mathbf{C}_i\}})
\end{aligned}
\tag{7}
$$

Above proof indicates that for a given generative model with any additional modal prior $\mathbf{C}_i$, its performance upper bound $\sup(\mathcal{P}_{\mathcal{C}_K \bigcup \{\mathbf{C}_i\}})$ is greater or equal to its original performance upper bound $\sup(\mathcal{P}_{\mathcal{C}_K})$. $\square$

**Theorem 2.** *If prior $\mathbf{C}_{pe}$ is a performance-enhancing prior, a generative model neglecting prior $\mathbf{C}_{pe}$ will decrease the upper bound of model performance. It can be expressed as*

$$
\sup(\mathcal{P}_{\mathcal{C} \setminus \{\mathbf{C}_{pe}\}}) < \sup(\mathcal{P}_{\mathcal{C}}) \tag{6}
$$

*Proof.* If prior $\mathbf{C}_{pe}$ is a performance-enhancing prior, according to Definition 2, we have

$$
\begin{aligned}
\max_{\mathbf{w}} p(\mathbf{w}|\mathbf{c}) \Phi(\mathbf{w})_{\mathbf{c} \in \mathcal{C} \setminus \{\mathbf{C}_{pe}\}} &< \max_{w} p(\mathbf{w}|\mathbf{c}) \Phi(\mathbf{w})_{\mathbf{c} \in \mathcal{C}} \\
&= \sum_{\mathbf{c}_{pe} \in \mathbf{C}_{pe}} p(\mathbf{c}_{pe}|\mathbf{c}) \max_{\mathbf{w}} p(\mathbf{w}|\mathbf{c}) \Phi(\mathbf{w})_{\mathbf{c} \in \mathcal{C}}
\end{aligned}
\tag{8}
$$

Then, according to Definition 1, $\sup(\mathcal{P}_{\mathcal{C} \setminus \{\mathbf{C}_{pe}\}})$ can be expanded as

$$
\sup(\mathcal{P}_{\mathcal{C} \setminus \{\mathbf{C}_{pe}\}}) = \sum_{\mathbf{C} \in \mathcal{C} \setminus \{\mathbf{C}_{pe}\}} \sum_{\mathbf{c} \in \mathbf{C}} p(\mathbf{c}) \max_{\mathbf{w}} p(\mathbf{w}|\mathbf{c}) \Phi(\mathbf{w}) \tag{9}
$$

Substitute Eq.(8) into Eq.(9) to derive

$$
\begin{aligned}
\sup(\mathcal{P}_{\mathcal{C} \setminus \{\mathbf{C}_{pe}\}}) &= \sum_{\mathbf{C} \in \mathcal{C} \setminus \{\mathbf{C}_{pe}\}} \sum_{\mathbf{c} \in \mathbf{C}} p(\mathbf{c}) \max_{\mathbf{w}} p(\mathbf{w}|\mathbf{c}) \Phi(\mathbf{w}) \\
&< \sum_{\mathbf{C} \in \mathcal{C} \setminus \{\mathbf{C}_{pe}\}} \sum_{\mathbf{c} \in \mathbf{C}} p(\mathbf{c}) \sum_{\mathbf{c}_{pe} \in \mathbf{C}_{pe}} p(\mathbf{c}_{pe}|\mathbf{c}) \max_{\mathbf{w}} p(\mathbf{w}|\mathbf{c}) \Phi(\mathbf{w}) \\
&= \sum_{\mathbf{C} \in \mathcal{C} \setminus \{\mathbf{C}_{pe}\}} \sum_{\mathbf{c} \in \mathbf{C}} \sum_{\mathbf{c}_{pe} \in \mathbf{C}_{pe}} p(\mathbf{c}) p(\mathbf{c}_{pe}|\mathbf{c}) \max_{\mathbf{w}} p(\mathbf{w}|\mathbf{c}) \Phi(\mathbf{w}) \\
&= \sum_{\mathbf{C} \in \mathcal{C} \setminus \{\mathbf{C}_{pe}\}} \sum_{\mathbf{c} \in \mathbf{C}} \sum_{\mathbf{c}_{pe} \in \mathbf{C}_{pe}} p(\mathbf{c}_{pe}, \mathbf{c}) \max_{\mathbf{w}} p(\mathbf{w}|\mathbf{c}) \Phi(\mathbf{w}) \\
&= \sum_{\mathbf{C} \in \mathcal{C}} \sum_{\mathbf{c} \in \mathbf{C}} p(\mathbf{c}) \max_{\mathbf{w}} p(\mathbf{w}|\mathbf{c}) \Phi(\mathbf{w}) \\
&= \sup(\mathcal{P}_{\mathcal{C}})
\end{aligned}
\tag{10}
$$

$\square$

# B Implementation Details

## B.1 Combinatorial Structure Extraction

**Data Representation of MILP problem.** A mixed-integer linear programming (MILP) problem is formally defined as:

$$\min_{\boldsymbol{x}\in\mathbb{R}^n} \boldsymbol{w}^\top \boldsymbol{x}, \quad \text{s.t. } \boldsymbol{A}\boldsymbol{x} \leq \boldsymbol{b}, \boldsymbol{l} \leq \boldsymbol{x} \leq \boldsymbol{u}, x_j \in \mathbb{Z}, \forall j \in \mathbb{I}, \tag{11}$$

where $\boldsymbol{w} \in \mathbb{R}^n$, $\boldsymbol{A} \in \mathbb{R}^{m \times n}$, $\boldsymbol{b} \in \mathbb{R}^m$, $\boldsymbol{l} \in (\mathbb{R} \cup \{-\infty\})^n$, $\boldsymbol{u} \in (\mathbb{R} \cup \{+\infty\})^n$, and the index set $\mathbb{I} \subset \{1, 2, \cdots, n\}$ specifies integer-constrained variables.

To encode MILP instances, we design a bipartite graph $\mathcal{G} = (\mathcal{C} \cup \mathcal{V}, \mathcal{E})$ with constraint-variable interactions. The constraint node set $\mathcal{C} = \{c_1, \cdots, c_m\}$ corresponds to rows of $\boldsymbol{A}\boldsymbol{x} \leq \boldsymbol{b}$, where each node $c_i$ is associated with a 1D feature vector $\boldsymbol{c}_i = (b_i)$ representing its right-hand side value. The variable node set $\mathcal{V} = \{v_1, \cdots, v_n\}$ represents decision variables, each equipped with a 9D feature vector $\boldsymbol{v}_j$ containing the objective coefficient $w_j$, variable type indicator (integer/continuous), and bound parameters $l_j, u_j$. Edges $\mathcal{E} = \{e_{ij}\}$ connect constraint $c_i$ to variable $v_j$ iff $a_{ij} \neq 0$, with edge features $\boldsymbol{e}_{ij} = (a_{ij})$ encoding the constraint coefficients.

Furthermore, a mixed-integer linear programming (MILP) instance is encoded as a weighted bipartite graph with feature matrices $\boldsymbol{G} = (\boldsymbol{C}, \boldsymbol{V}, \boldsymbol{E})$, where $\boldsymbol{C}, \boldsymbol{V}$, and $\boldsymbol{E}$ aggregate constraint node features $\boldsymbol{c}_i$, variable node features $\boldsymbol{v}_j$, and edge features $\boldsymbol{e}_{ij}$, respectively. The full specification of these features is summarized in Table 2, which preserves all structural and numerical information of the original MILP problem. Following standard practice, we utilize the observation function from Ecole [50] to generate these bipartite graph representations from MILP instances.

Table 2: Description of the constraint, variable, and edge features in our bipartite graph representation for MILP instance.

| Tensor | Feature | Description |
|---|---|---|
| $\boldsymbol{C}$ | Constraint coefficient | Average of all coefficients in the constraint. |
| | Constraint degree | Degree of constraint nodes. |
| | Bias | Normalized right-hand-side of the constraint. |
| $\boldsymbol{V}$ | Objective | Normalized objective coefficient. |
| | Variable coefficient | Average variable coefficient in all constraints. |
| | Variable degree | Degree of the variable node in the bipartite graph representation. |
| | Maximum variable coefficient | Maximum variable coefficient in all constraints. |
| | Minimum variable coefficient | Minimum variable coefficient in all constraints. |
| $\boldsymbol{E}$ | Coefficient | Constraint coefficient. |

**Data Representation of SAT problem.** A Boolean satisfiability (SAT) problem consists of variables $x_i$ and logical operators $\wedge, \vee$, and $\neg$. A formula is satisfiable if there exists a variable assignment that makes all clauses evaluate to true. Following [51], we focus on formulas in conjunctive normal form (CNF) – conjunctions ($\wedge$) of clauses, where each clause is a disjunction ($\vee$) of literals (variables $x_i$ or their negations $\neg x_i$). Any SAT formula can be converted to an equisatisfiable CNF formula in linear time. For example, $(x_1 \vee \neg x_2) \wedge (x_2 \vee \neg x_3)$ represents a CNF formula with two clauses.

We utilize the Variable-Clause Graph (VCG) to represent SAT formulas. For a SAT formula, the VCG is constructed with nodes representing literals and clauses, and edges indicating the inclusion of a literal in a clause. The bipartite structure of VCGs ensures a one-to-one correspondence between CNF formulas and their graph representations. Formally, a bipartite graph $\mathcal{G} = (\mathcal{V}_1 \cup \mathcal{V}_2, \mathcal{E})$ is defined by its vertex set $\mathcal{V}_1 \cup \mathcal{V}_2 = \{v_1, ..., v_n\}$ and edge set $\mathcal{E} \subseteq \{(v_i, v_j) | v_i, v_j \in \mathcal{V}\}$. The vertex set is partitioned into two disjoint subsets $\mathcal{V}_1$ and $\mathcal{V}_2$, with edges restricted to connections between nodes in distinct partitions: $\mathcal{E} \subseteq \{(v_i, v_j) | v_i \in \mathcal{V}_1, v_j \in \mathcal{V}_2\}$. In the context of VCGs, a CNF formula with $n$

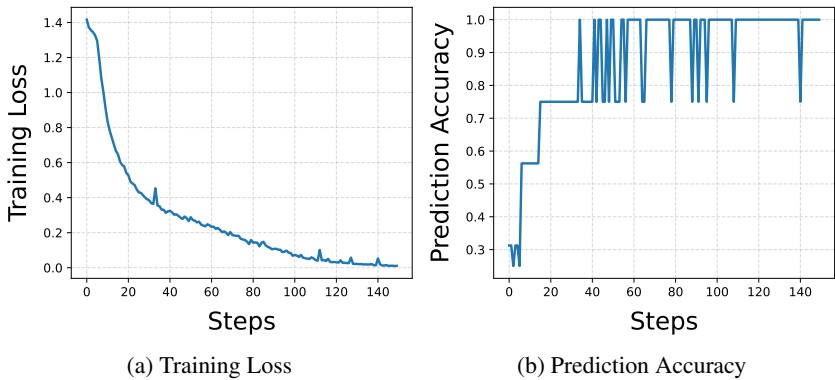

(a) Training Loss

(b) Prediction Accuracy

Figure 6: The convergence curve of training GNN for SAT domain.

literals and $m$ clauses induces a bipartitioned graph where $\mathcal{V}_1 = \{l_1, ..., l_n\}$ denotes the literal nodes and $\mathcal{V}_2 = \{c_1, ..., c_m\}$ represents the clause nodes.

**GNN Structure.** The process of using GNN to extract combinatorial optimization problem can be formulated as:

$$\mathbf{v}_i^{(k+1)} \leftarrow \mathbf{f_V}(\mathbf{v}_i^{(k)}, \sum_{j, i \neq j}^{(i,j) \in \mathcal{E}} \mathbf{g_V}(\mathbf{v}_i^{(k)}, \mathbf{v}_j^{(k)}, \mathbf{e}_{ij})), \quad (k = 0, 1, ..., K - 1) \tag{12}$$

$$\mathbf{h}_q = Pool(\{\mathbf{v}_i^{(K)}\}), \tag{13}$$

where $\mathbf{f_V}$ and $\mathbf{g_V}$ are perceptrons for node representation; $K$ represents the total number of times that we perform the convolution; $Pool$ denotes pooling function that aggregates the embedding of each node in the graph, obtaining the structure embedding $\mathbf{h}_q \in \mathbb{R}^d$ of the instance $q$. We denote the parameters of GNN as $\theta_G$.

**GNN Training.** The loss function w.r.t. $\theta_G$ is given below:

$$\mathcal{L}(\theta_G) = -\frac{1}{N} \sum_{i=1}^{N} \sum_{c=1}^{C} y_{i,c} \log p_{\theta_G}(c|q_i), \tag{14}$$

$$p_{\theta_G}(c|q_i) = \frac{\exp\left(\mathbf{W}_c^T \mathbf{h}_{q_i} + \mathbf{b}_c\right)}{\sum_{j=1}^{C} \exp\left(\mathbf{W}_j^T \mathbf{h}_{q_i} + \mathbf{b}_j\right)}, \tag{15}$$

where $N$ is the total number of CO problems in the training procedure; $C$ denotes the number of classes; $y_{i,c} \in \{0, 1\}$ is the ground-truth label of problem instance $q_i$; and $\mathbf{W}_j \in \mathbb{R}^d, \mathbf{b}_j \in \mathbb{R}, (j = 1, ..., C)$ is the parameters of final classifying layer, which are also part of $\theta_G$.

**Implementation & Training Details.** Following the training protocol outlined above, we implement separate GNN models for the SAT and MILP domains using the dataset described in Appendix D. Our GNN architecture consists of three convolutional layers ($K = 3$) followed by a global mean pooling operation. We leverage the graph convolution operator from the `torch_geometric` library, with node embedding dimensions of 16, 32, and 64 for successive layers[3]. To capture global graph structure, we apply mean pooling to the final convolutional layer's node embeddings. For classification, a softmax layer processes the pooled representation to produce final predictions. The models are trained using the loss function defined in Eq.(14) with AdamW optimizer and cosine decay learning rate. Figure 6 illustrates the training convergence for SAT instance classification (5-way classification task). Upon model convergence, we compute structural prior representations for combinatorial optimization problem instances by processing their bipartite graph representations through the above GNN. The final convolutional layer's output embeddings are extracted as the structural prior for each instance in the target domain.

---

[3]For the MILP domain, we increase the final layer's dimensionality to 128 to account for higher problem complexity compared to SAT domain.

## B.2 Structure-Aware Code Generation

**Data Curation.** We begin by assembling a curated collection of mathematical models for CO problems, which are directly compatible with corresponding CO solvers, alongside their natural language descriptions. Subsequently, for each CO problem, we compile a prompt that integrates the natural language description with specific code generation requirements. For instance, the code generation requirements may include details such as the function name, input/output parameters, expected function behavior, relevant background knowledge, *etc*. Illustrative examples for SAT and MILP domain are provided in Appendix L. This prompt is then fed into an LLM to generate a code snippet. Note that to maximize the diversity of collected code snippets, we employ multiple LLM queries per prompt under high temperature settings to sample distinct candidate implementations. Each generated code snippet is evaluated by embedding it into the target solver and solving the corresponding CO problem, thereby obtaining performance metrics for the code snippet. Note that based on the principles of data curation, we collect the needed data via querying `Qwen2.5-Coder-7B-Instructor` same as the model adopted in the following post training procedure.

**Post Training.** Specifically, we first curate the collected data by processing each CO problem $Q_i$ with its associated prompt $x_i$ and corresponding multiple generated code snippets $y_{i,j}(j = 1, ..., M)$. These code snippets are systematically ranked based on previously obtained performance metrics to establish quality ordering. The highest-performing code-prompt pairs are selected to form the SFT dataset $\mathcal{D}_{SFT} = \{(x_i, y_i^*)\}_{i=1}^N$. Additionally, by leveraging pairwise comparisons extracted from the established ranking hierarchy, we derive the preference dataset $\mathcal{D}_{DPO} = \{(x_i, y_w, y_l)\}_{i=1}^N$ where $y_w, y_l$ are preferred/dispreferred code snippets. The training process can be formulated as follows:

$$\mathcal{L}_{\text{DPO}} = -\mathbb{E}_{(x,y_w,y_l)\sim\mathcal{D}_{DPO}} \left[ \log \sigma \left( \beta \ln \frac{\pi_{\theta_L}(y_w|x)}{\pi_{\text{ref}}(y_w|x)} - \beta \ln \frac{\pi_{\theta_L}(y_l|x)}{\pi_{\text{ref}}(y_l|x)} \right) \right], \quad (16)$$

where $\sigma$ is the Sigmoid function; $\beta$ is the hyperparameter that governs the trade-off between reward maximization and KL divergence minimization; $\pi_{\text{ref}}$ is the reference model trained on $\mathcal{D}_{SFT}$. Note that both $\pi_{\text{ref}}$ and $\pi_{\theta_L}$ updates only the parameter $\theta_L$ of the composite model. Through above post-training, we obtain a generative model capable of structure-aware code generation that simultaneously respects combinatorial optimization problems' inherent topological constraints and solver-specific syntactic requirements. Note that based on above principle of data curation and post-training, we collect 8k and 4k post-training instances for MILP and SAT instances respectively.

**Implementation & Training Details.** To implement the proposed composite model, we first design a structure-prior-aware forward propagation mechanism and corresponding adapted inference framework based on the `Qwen2.5-Coder-7B-Instructor`[4] model via the `Transformers` library.

❶ **Structure-Prior-Aware Forward Propagation Mechanism.** Specifically, we process the input prompt through the tokenizer and the model's embedding layer to obtain $input\_embeds$, which are then merged with structural feature vectors of combinatorial optimization problems extracted by the previous graph neural network.

First, we align the dimensionality of the combinatorial optimization problem feature vector $\boldsymbol{h}_q \in \mathbb{R}^d$ with the hidden layer dimensions of the large language model via zero-padding. Given the text embedding shape $embeds \in \mathbb{R}^{B \times S \times H}$, where $B$ is the batch size, $S$ is the sequence length (number of tokens), and $H$ is the hidden dimension, the dimension-adapted feature vector is obtained as:

$$\mathbf{H}_q = ZeroPadding(\boldsymbol{h}_q \oplus \mathbf{0}^{(H-d)}) \in \mathbb{R}^{B \times H} \quad (17)$$

where $ZeroPadding$ denotes zero-padding, and $\mathbf{H}_q$ represents the padded graph structural feature vector. Subsequently, we fuse text and structural features between the embedding layer and decoder layer by prepending the graph feature vector to the text embedding sequence, forming a hybrid input:

$$\mathcal{E}(embeds, \mathbf{H}_q) = [\text{CLS}] \oplus \mathbf{H}_q \oplus embeds[1:] \in \mathbb{R}^{B \times (1+S) \times H} \quad (18)$$

Here, $embeds$ is the $input\_embeds$ obtained from processing the textual prompt via the tokenizer and embedding layer, enabling the self-attention mechanism to jointly model textual semantics and graph structural features. A mask of all True values is constructed and merged with the $attention\_mask$ to match the shape of the combined $input\_embeds$, ensuring $\mathbf{H}_q$ participates in attention computation. Let the original attention_mask shape be $M = \{0, 1\}^{B \times S}$; the merged $attention\_mask$ becomes:

$$M_{in} = [M_{0:1}; \mathbf{1}^B; M_{1:S}] \in \{0, 1\}^{B \times (1+S)} \quad (19)$$

---

[4]This model is available at https://huggingface.co/Qwen/Qwen2.5-Coder-7B-Instruct.

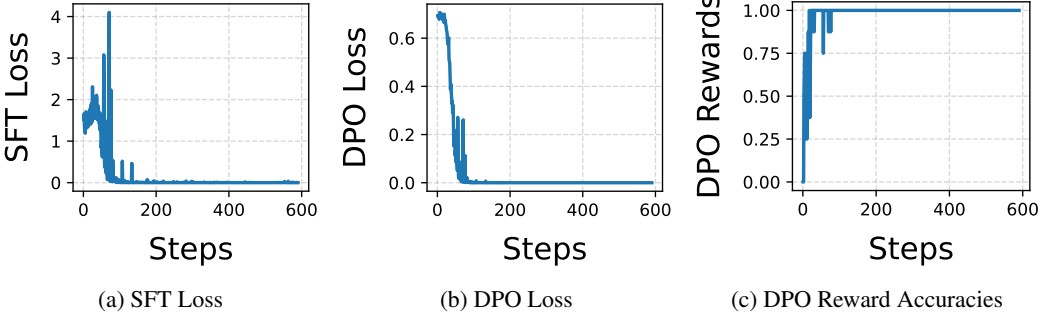

(a) SFT Loss       (b) DPO Loss       (c) DPO Reward Accuracies

Figure 7: The convergence curve of post-training composite model for SAT domain.

The resulting $input\_embeds$ and $attention\_mask$ are then passed to the decoder layers and subsequent model structures for standard forward propagation.

❷ **Parameter-Efficient Finetuning.** We adopt LoRA for parameter-efficient fine-tuning on an autoregressive language model task. Key hyperparameters are configured as: rank dimension $r = 16$ controlling the latent dimension of low-rank adaptation matrices; scaling factor $lora\_alpha = 32$ for output normalization; dropout probability $0.05$ for regularization; trainable low-rank adaptation layers injected exclusively into query ($q\_proj$) and value ($v\_proj$) projection submodules of the Transformer architecture while keeping other parameters frozen; bias parameters set to "none" to preserve original model biases. The adapted LLM is trained with AdamW optimizer and cosine decay learning rate. An illustrative training curve of the model for SAT domain is given in Figure 7.

❸ **Adapted Inference Framework.** Based on this modified architecture, we further adapt the LLM inference framework with structure-prior feature vector integration. During each autoregressive forward pass, logits are obtained via the feature-enhanced forward propagation, ensuring persistent influence of the structure-prior feature vector throughout the generation process rather than only affecting the initial output. We employ a hybrid sampling strategy with Top-k set to 20, Top-p to 0.8, and repetition penalty to 1.0.

## C   Baseline Details

### C.1   Solver Adoption

**MILP Solver.** In all experiments related to MILP domain, we employ SCIP 8.0.0 [52] as the MILP solver backend—a leading open-source solver widely adopted in machine learning research for combinatorial optimization [44, 53]. To ensure reproducibility and fair comparisons, all SCIP parameters remain at default values. We retain the solver's advanced features (*e.g.*, presolve, heuristics), aligning our experimental setup with real-world applications.

**SAT Solver.** In all experiments related to SAT domain, we use EasySAT[5] to ensure direct comparability with AutoSAT[29]. The solver incorporates modern Conflict-Driven Clause Learning (CDCL) techniques including Literal Block Distance (LBD) heuristics, and VSIDS variable selection, and conflict-driven clause learning. All configurations (compiler versions, interfaces, time budgets) maintain parity with [29] for reproducibility.

### C.2   Baselines Setting

#### C.2.1   Neural Combinatorial Optimization

**L2B [14].** The paper addresses the challenge of variable selection in the branch-and-bound (B&B) algorithm for solving mixed-integer linear programs (MILPs), where traditional expert-designed heuristics like strong branching incur prohibitive computational costs. To overcome this, the authors propose a graph convolutional neural network (GCNN) framework that leverages the natural bipartite

---

[5]EasySAT: A Simple CDCL SAT Solver, https://github.com/shaowei-cai-group/EasySAT.

graph representation of MILPs – with variable and constraint nodes connected via edges representing their coefficients – to learn branching policies through imitation learning. Key innovations include encoding MILP states as bipartite graphs with constraint/variable features, designing permutation-invariant sum-based graph convolutions with prenormalization layers to handle variable-sized inputs, and training via behavioral cloning of strong branching decisions using cross-entropy loss. All configurations take the default value used in [14] for fair comparison.

**HEM [15, 16].** The paper addresses the challenge of improving cut selection in mixed-integer linear programming (MILP) solvers by simultaneously optimizing three critical aspects: which cuts to select (P1), how many to choose (P2), and their ordering (P3). Existing learning-based methods focus primarily on scoring individual cuts while neglecting dynamic cut count determination and sequence dependencies. To overcome these limitations, the authors propose a novel hierarchical sequence model (HEM) that leverages a two-level architecture: a higher-level policy predicts the ratio of cuts to select using a tanh-Gaussian distribution, while a lower-level pointer network formulates cut selection as a sequence-to-sequence learning problem to output ordered subsets. The model is trained via hierarchical policy gradient optimization within a reinforcement learning framework, where states encode MILP relaxation features and candidate cut characteristics, actions represent ordered cut subsets, and rewards correspond to solver performance metrics like primal-dual gap integral. All hyperparameters take the default value used in [15, 16] for fair comparison.

**NeuroSAT [40].** The paper proposes a message-passing neural network (MPNN) to solve the Boolean satisfiability problem by training solely on binary labels indicating satisfiability. The key innovation lies in representing SAT instances as bipartite graphs where literals and clauses are nodes connected via edges, and leveraging permutation-invariant neural architectures to process these graphs. NeuroSAT iteratively refines node embeddings through bidirectional message passing: clauses aggregate information from their constituent literals, while literals update their states based on connected clauses and complementary literals. Trained on a distribution of random SAT problems generated by incrementally adding clauses until unsat, the model learns to predict satisfiability via a cross-entropy loss on the final aggregated literal embeddings. Crucially, the architecture enforces structural symmetries (variable/clause permutation invariance, negation equivalence) and generalizes to larger instances and entirely unseen domains (*e.g.*, graph coloring, vertex cover) at test time by simply extending the message-passing iterations, despite being trained only on small random $n \leq 40$ problems. All configurations of this algorithm take the default used in [40] for fair comparison. Note that since NeuroSAT is a prediction framework for Boolean satisfiability problem, it is meaningless to measure the solving time for this framework over the SAT instances. Besides, the ground-truth dataset used to train the NeuroSAT model is those SAT instances can be proved/solved within timelimit $\tau$. Thus, we only count the number predicted to be correct of NeuroSAT (this number must be less than the number of ground-truth labels). Then, the number of timeout for NeuroSAT is equal to the total number of test instances minus the number predicted to be correct.

### C.2.2 Evolutionary Code Optimization

**AutoSAT [29].** The paper introduces a framework that leverages Large Language Models (LLMs) to automatically optimize heuristics in Conflict-Driven Clause Learning (CDCL) SAT solvers. The authors address the challenge of manually designing and tuning heuristic functions in modern CDCL solvers, which is time-consuming and expert-dependent. Instead of generating solvers from scratch, AutoSAT operates within a modular search space comprising nine key heuristic functions (*e.g.*, branching, restart, and clause management) derived from an existing CDCL solver. The framework employs LLMs to iteratively refine these heuristics through two search strategies: a greedy hill climber (GHC) and a (1+1) Evolutionary Algorithm (EA), where LLMs generate candidate code modifications guided by performance feedback. All configurations of this algorithm take the default used in [29] for fair comparison.

**LLM4Solver [30].** The paper proposes a novel framework that integrates large language models (LLMs) with evolutionary search to automate the design of high-performance diving heuristics for exact combinatorial optimization (CO) solvers. The key challenge addressed is the inefficiency of traditional manual and learning-based approaches in navigating the vast, discrete algorithm space for CO solver components like diving heuristics, which require domain expertise and suffer from poor generalization. The core methodology leverages LLMs as prior-knowledge-guided generators to produce candidate algorithms encoded as interpretable score functions, while a derivative-free evolutionary framework (single- or multi-objective) optimizes these candidates through iterative

population-based search. The algorithm space is defined via 13 interpretable variable features, with LLMs enabling code-level manipulations during initialization, crossover, and mutation. We slightly adjusted **LLM4Solver**'s optimization focus from diving heuristics to cut selection for MILP solver for fair comparison with previous work [15, 16]. All configurations of this algorithm take the default used in [15, 16] for fair comparison.

## D  Dataset Details

### D.1  SAT Domain

**Chromatic-Number-of-the-Plane (CNP).** This problem requires coloring graph vertices such that adjacent nodes have distinct colors.

**Profitable-Robust-Production (PRP).** This problem seeks robust production plans maintaining profitability under uncertain/fluctuating conditions. The generation setting of PRP adhere to [29].

**CoinsGrid.** This problem, based on Tony Hurlimann's coin puzzle, involves arranging coins on a grid with row-wise, column-wise, and distance-based constraints. The generation setting of CoinsGrid adhere to [29].

**Zamkeller.** This problem requires finding a permutation of integers from $1$ to $n$ that maximizes differential alternations in subsequences divisible by integers $1$ to $k$ ($1 < k < n$). Instance generation follows [29].

Table 3: The statistical description of used SAT datasets. Const. and Var. stand for constraints and variables respectively.

| Dataset | Size | # Const. (mean) | # Const. (std) | # Var. (mean) | # Var. (std) |
|---------|------|-----------------|----------------|---------------|--------------|
| CNP | 150 | 261,260 | 352,379 | 8,798 | 8,893 |
| CoinsGrid | 78 | 1,116,972 | 1,068,009 | 154,828 | 148,571 |
| PRP | 80 | 2,120,983 | 1,346,344 | 317,635 | 201,185 |
| Zamkeller | 80 | 310,804 | 335,102 | 24,592 | 22,190 |

### D.2  MILP Domain

**Easy.** The SCIP 8.0.0 solver solves MILP instances in the Easy dataset within one minute. This dataset comprises three synthetic problems: Set Covering [41], Maximum Independent Set [42], and Multiple Knapsack [43]. We select these classes because: (1) they serve as standard benchmarks for MILP solver evaluation [44], and (2) they encompass diverse MILP problem structures encountered in practice. Following [44, 43], we generate set covering instances (500 rows × 1000 columns), Maximum Independent Set instances (500-node graphs with affinity 4), and multiple knapsack instances (60 items, 12 knapsacks). For each dataset within this category, we generate 1,000 training instances and hold 100 instances as test set. Besides, we set the timelimit as 60 seconds for solving problem instances within this dataset.

**Medium.** The SCIP 8.0.0 solver requires at least five minutes to solve this set of instances optimally. Following [54, 55, 53], this dataset combines MIK[6] [46] (MILPs with knapsack constraints) and CORLAT[7] [47] (real-world grizzly bear corridor planning in Northern Rockies). Each problem set is partitioned into 80% training and 20% test instances. Furthermore, we set the timelimit as 120 seconds for solving problem instances within this dataset.

**Hard.** The SCIP 8.0.0 solver requires more than one hour to solve Hard dataset instances to optimality. These datasets[8] include Load Balancing from the NeurIPS 2021 ML4CO competition [48]. Load Balancing instances model server task allocation under resource constraints. Same as Medium dataset, each problem set in Hard dataset is also partitioned into 80% training and 20% test instances. Additionally, we set the timelimit as 360 seconds for solving problem instances within this dataset.

---

[6]MIK data can be found at https://atamturk.ieor.berkeley.edu/data/mixed.integer.knapsack/.

[7]CORLAT data is available at https://bitbucket.org/mlindauer/aclib2/src/master/.

[8]All these data can be found at https://www.ecole.ai/2021/ml4co-competition/.

Table 4: The statistical description of used MILP datasets.

| Dataset | # Constraints (mean) | # Varaibles (mean) |
|---|---|---|
| Set Covering (SC) | 500 | 1,000 |
| Maximum Independent Set (MIS) | 1,953 | 500 |
| Knapsack | 72 | 720 |
| MIK | 346 | 413 |
| CORLAT | 486 | 466 |
| Load Balancing | 64,304 | 61,000 |

# E    Metric Details

## E.1    Metrics related to MILP

For fair comparison, our evaluation employs metrics aligned with HEM [15, 16, 14], which are well-established in the MILP community.

**Solving Time.** For MILP instances solved to optimality within time limit $T$, we measure the solver runtime $t$ required to obtain certifiably optimal solutions. For those MILP instances that cannot be solved to optimality within time limit $T$, we utilize the following metrics.

**Primal-Dual Gap.** During the execution of MILP solvers, two critical bounds are maintained: the global primal bound and the global dual bound. The global primal bound represents the objective value of the incumbent feasible solution, serving as the tightest upper bound for the MILP. The global dual bound corresponds to the minimal lower bound among all active nodes in the search tree, constituting the strongest lower bound for the problem. The primal-dual gap is defined as the absolute difference between these two bounds. In SCIP 8.0.0 [52], the initial primal-dual gap is initialized to a constant value of 100.

**Primal-Dual (PD) Integral.** The primal-dual gap integral is defined as the area enclosed between the primal and dual bound trajectories during solver execution. Formally, given a time limit $T$, this integral is expressed as:

$$\int_{t=0}^{T} (\boldsymbol{w}^T \boldsymbol{x}_t^* - \mathbf{z}_t^*) \mathrm{d}t, \tag{20}$$

where $\mathbf{w}$ denotes the objective coefficient vector of the MILP instance; $\boldsymbol{x}_t^*$ represents the incumbent feasible solution at time $t$, $\mathbf{z}_t^*$ corresponds to the tightest dual bound at time $t$. This metric quantifies the cumulative optimality gap over the solving horizon and serves as an established performance benchmark for MILP solvers. Notably, the primal-dual gap integral was adopted as the primary evaluation criterion in the NeurIPS 2021 ML4CO competition [48].

## E.2    Metrics related to SAT

To ensure fair comparison, our evaluation adopts metrics consistent with AutoSAT [29], which are widely recognized in the SAT community.

**Solving Time.** For SAT instances that can be solved/proved within a timeout limitation $\tau$, we record the running time $t$ of a SAT solver solving/proving these instances. For those SAT instances that cannot be solved/proved within timeout limitation $\tau$, we measure the two following metrics.

**Number of Timeout.** Given a timeout limitation $\tau$ and a set of SAT instances, we record the number of SAT instances that cannot be solved/proved within $\tau$ for different compared methods. The $\tau$ is set as 5000 seconds, same as [29].

**Penalized Average Runtime with a factor of 2 score (PAR-2).** PAR-2 is a standard evaluation metric in SAT solver competitions and benchmarks, employed to assess and compare solver performance under incomplete executions or timeout scenarios. Specifically, the PAR-2 score aggregates runtime across test instances as follows: 1) For solved instances, record the solver's runtime; 2) For instances where the solver exceeds the timeout limitation $\tau$, assign twice the timeout threshold $\tau$ as the penalty—this is why we call this metric as PAR-2; 3) Compute the average across all instances by summing these values and dividing by the instance count. Lower PAR-2 scores indicate better overall solver efficiency.

Intuitively, consider a benchmark set comprising $n$ SAT instances. Let $t_i$ denote the solver's runtime on the i-th instance. The PAR-2 score is formally defined as

$$\text{PAR-2} = \frac{1}{n} \sum_{i=1}^{n} t_i, \tag{21}$$

where

$$t_i = \begin{cases} t_i, & \text{if } t_i \leq \tau; \\ 2\tau, & \text{if } t_i > \tau \text{ or the solver fails to return a solution.} \end{cases} \tag{22}$$

## F Discussion

**End-to-End Training.** In this work, we adopt a two-stage training procedure for our composite model: first training a graph neural network (GNN), then training a large language model (LLM) conditioned on the frozen GNN embeddings. This approach stems from our empirical observation that end-to-end joint training of the composite model presents significant optimization challenges. We hypothesize that this limitation could be mitigated by exploring more sophisticated GNN architectures or developing enhanced alignment strategies between the GNN's structural representations and the LLM's semantic space.

**Explore Domain-specific Distributional Discrepancies.** In our ablation studies, empirical results reveal that the full STRCMP model (with complete post-training) underperforms relative to its variants STRCMP (SFT Only) and STRCMP (DPO Only) across benchmark datasets, suggesting potential conflicts between optimization objectives during multi-stage post-training. This observation motivates future investigation into domain-specific distributional discrepancies that may arise from the heterogeneous nature of combinatorial optimization problems, which could inform improved alignment strategies for cross-domain post-training protocols.

**Interpretability of the generated code functions scale with the increasing complexity of the CO problem.** We fully acknowledge that while code generation enhances transparency relative to neural black-box methods, interpretability does not scale linearly with problem complexity. For highly intricate CO problems or novel heuristics, generated code may involve nested logic, context-dependent optimizations, or non-intuitive transformations. This necessitates non-trivial effort for human experts to parse, reducing immediate interpretability. Nevertheless, STRCMP retains fundamental advantages over neural NCO methods: 1) Inspectability: solver logic remains open to direct inspection in its artifactual form; 2) Modifiability: code can be edited without model retraining; and 3) Debuggability: runtime validation and issue tracing via standard tools remain possible. We view the interpretability of complex generated code as an ongoing research challenge and are committed to addressing it in future work. Potential directions include simplifying the generated algorithms through post-processing or enhancing them with annotations to aid human understanding. For now, we believe STRCMP strikes a valuable balance, advancing interpretability over NCO methods while laying the groundwork for further improvements.

**Incorporating Additional Modal Priors.** In this study, we focus specifically on integrating graph structural priors into LLM-based algorithm discovery for combinatorial optimization problems. While current methodologies rely on human expertise to predefine target components, we posit that LLMs' context-aware capabilities could assimilate additional modal priors (e.g., dynamic constraint patterns or solution quality metrics) to enhance combinatorial optimization problem-solving performance. Future extensions may enable LLMs to autonomously identify computational bottlenecks through techniques such as runtime complexity profiling or constraint violation pattern analysis, thereby prioritizing component optimization.

# G    Additional Experimental Results

## G.1    Optimization Performance Result

Table 5: The optimization performance result w.r.t. PAR-2 score between different methods over SAT domain.

| Compared Methods | PAR-2 Score (↓) | | | |
|---|---|---|---|---|
| | CNP | CoinsGrid | PRP | Zamkeller |
| AutoSAT | 644.72 | 1098.69 | 639.34 | 807.75 |
| EasySAT | 649.42 | 1595.75 | 2000 | 829.38 |
| STRCMP | **624.15** | 1124.09 | 1804.46 | 270.62 |
| STRCMP (DPO Only) | 643.45 | 1098.69 | **482.92** | 265.94 |
| STRCMP (SFT Only) | 643.23 | 1098.47 | 1837.86 | **227.69** |
| STRCMP w/o GNN | 645.67 | **1098.34** | 1820.38 | 646.12 |

Table 6: The optimization performance result w.r.t. Solving Time between different methods over SAT domain.

| Compared Methods | Solving Time (↓) | | | |
|---|---|---|---|---|
| | CNP | CoinsGrid | PRP | Zamkeller |
| AutoSAT | 32472 | 19158 | 22967 | 20772 |
| EasySAT | 32942 | 26064 | 50000 | 21810 |
| STRCMP | **31415** | **18971** | 46223 | 7990 |
| STRCMP (DPO Only) | 32345 | 19158 | **21146** | 7765 |
| STRCMP (SFT Only) | 32323 | 19151 | 46893 | **6929** |
| STRCMP w/o GNN | 32567 | 19147 | 47019 | 17014 |

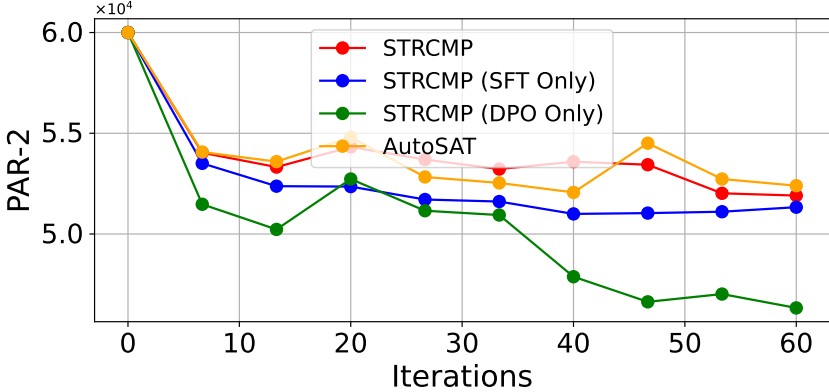

Figure 8: Convergence comparison (w.r.t. PAR-2) between evolutionary-based algorithm discovery frameworks on PRP dataset of SAT domain.

## G.2 Convergence Comparison Result

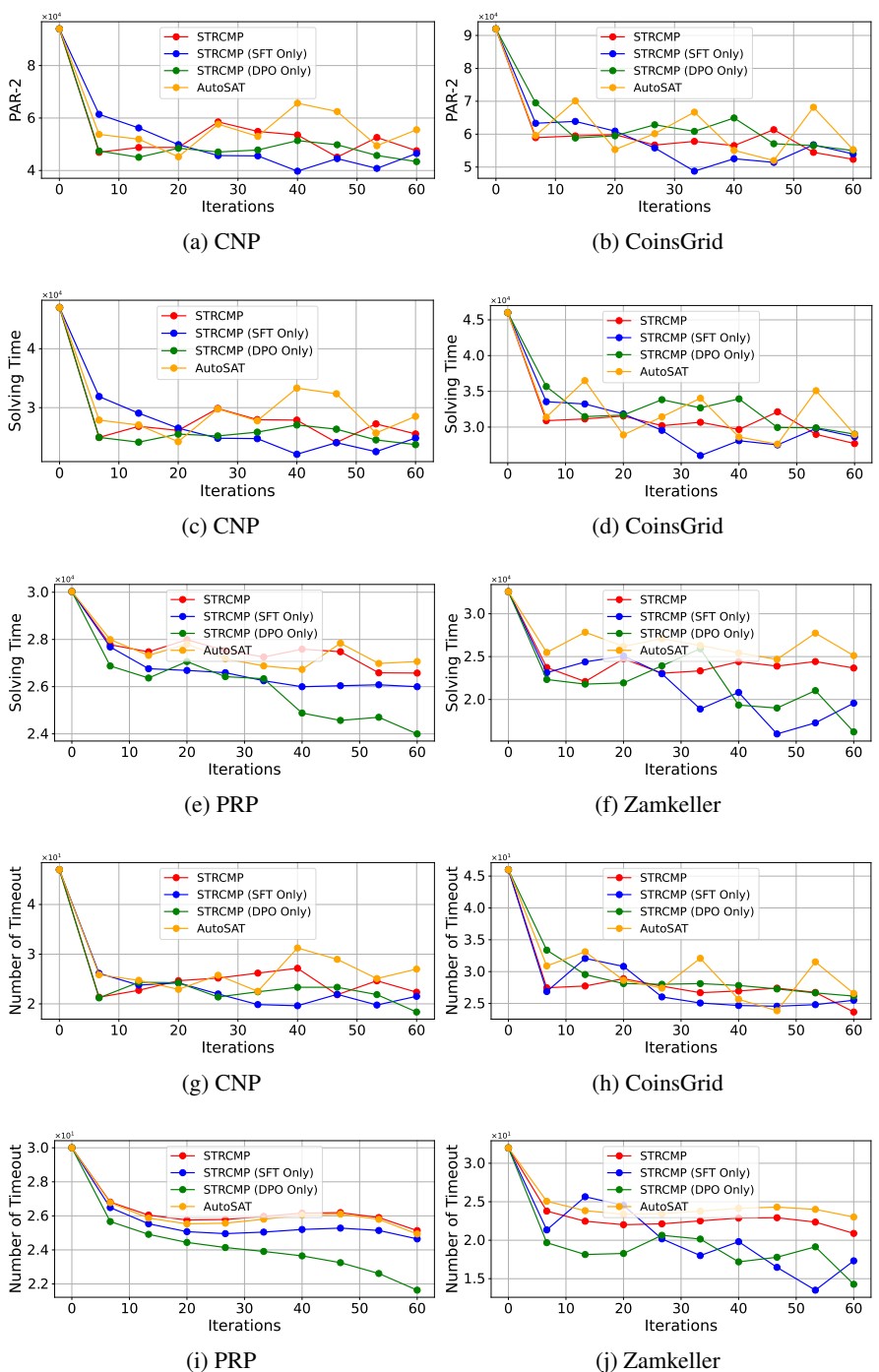

Figure 9: Convergence comparison (w.r.t. PAR-2, Solving Time, and Number of Timeout) between evolutionary-based algorithm discovery frameworks on SAT domain.

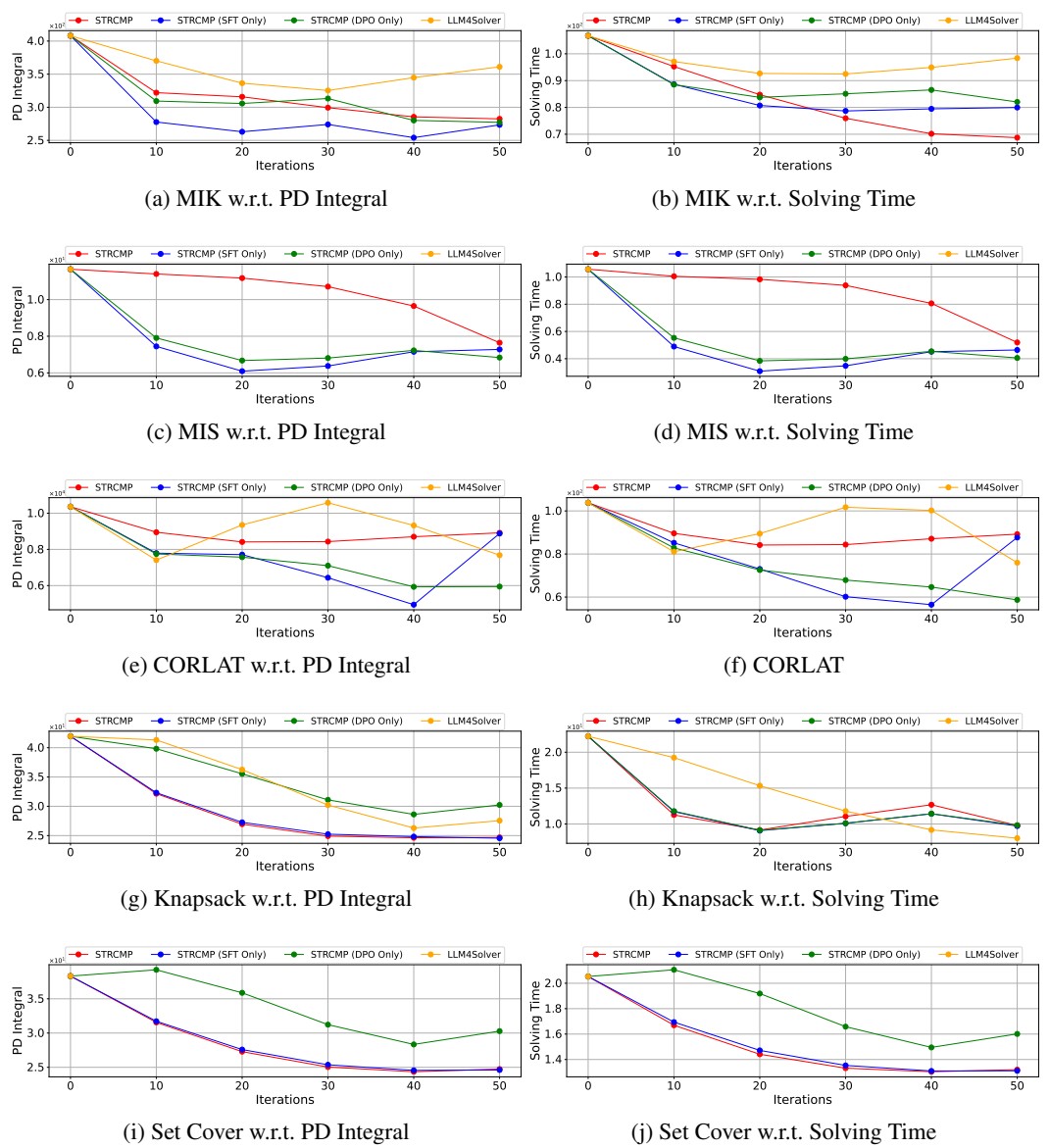

(a) MIK w.r.t. PD Integral

(b) MIK w.r.t. Solving Time

(c) MIS w.r.t. PD Integral

(d) MIS w.r.t. Solving Time

(e) CORLAT w.r.t. PD Integral

(f) CORLAT

(g) Knapsack w.r.t. PD Integral

(h) Knapsack w.r.t. Solving Time

(i) Set Cover w.r.t. PD Integral

(j) Set Cover w.r.t. Solving Time

Figure 10: Convergence comparison (w.r.t. PD Integral and Solving Time) between evolutionary-based algorithm discovery frameworks on MILP domain.

## G.3 Ablation Studies Result

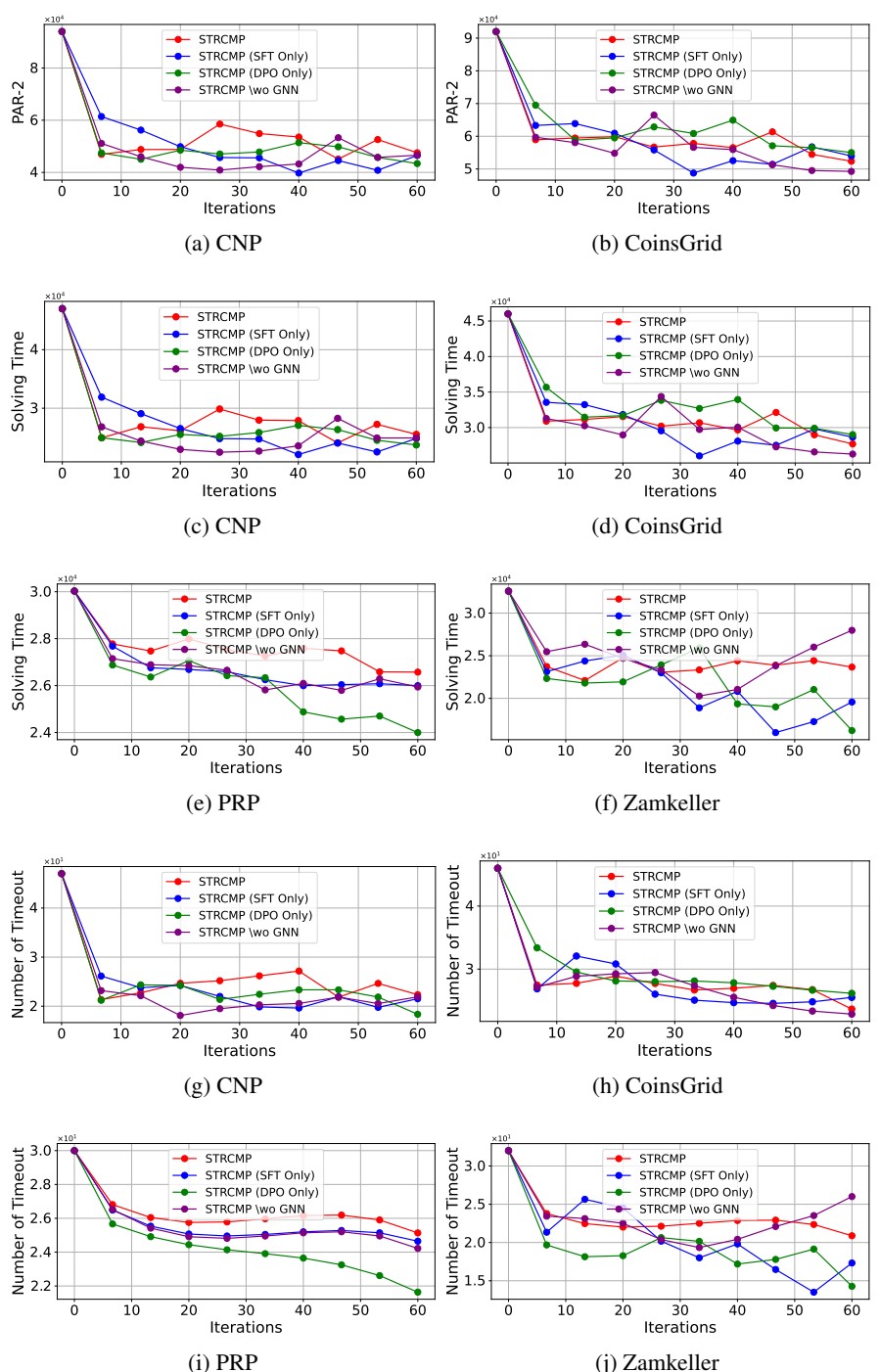

Figure 11: Ablation studies (w.r.t. PAR-2, Solving Time, and Number of Timeout) during algorithm search on SAT domain.

# H    Experiment Statistical Significance

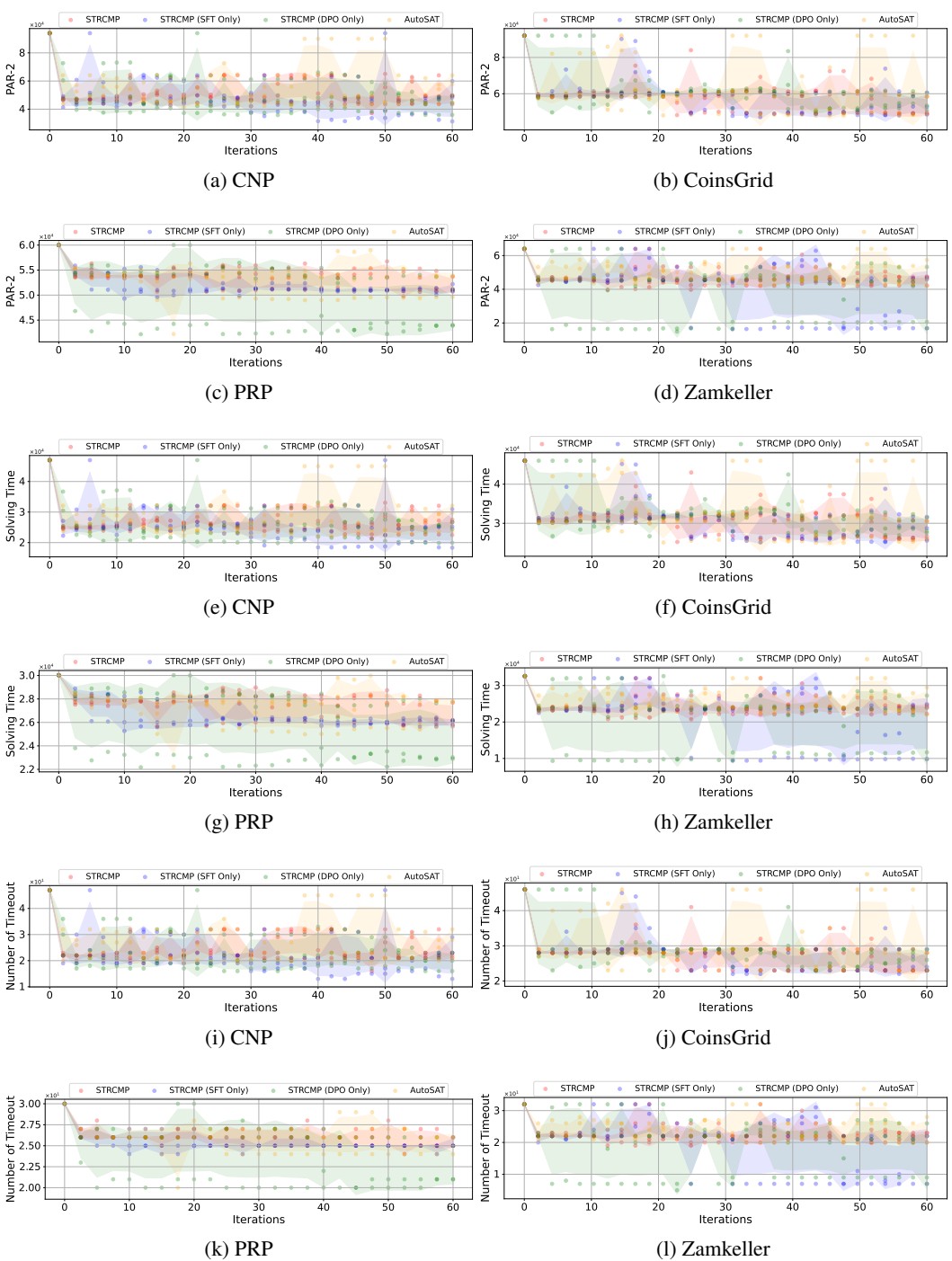

Figure 12: Convergence comparison with variance statistic (w.r.t. PAR-2, Solving Time, Number of Timeout) between evolutionary-based algorithm discovery frameworks on SAT domain.

# I Comparison between Two-stage training and Joint End-to-End Training

The detailed results of this new training and testing are presented show in Table 7 and Table 8. We can conclude the followings from these results:

❶ **The end-to-end joint training is extremely unstable compared to our proposed two-stage training method**. We infer that the training instability stems from the architectural and optimization differences between the GNN and the LLM. The gradients, originating from a sequence-level generation loss in the deep LLM, become noisy and attenuated when backpropagated through the entire LLM and into the much shallower GNN. This creates a challenging optimization landscape where a single learning rate is ineffective for both components, leading to unstable convergence (even with many more training steps) as shown in Table 7. Compared with the joint training, the loss curve of our two-stage training method is stable and close to zero at convergence (Please see the loss curve of our two-stage training method in Figure 7 in Appendix). The GNN and LLM are trained together, which makes it difficult for the models to learn optimal parameters simultaneously.

❷ **The model exhibits poor performance when tested on solving SAT instances**. Unsurprisingly, due to its very unstable training and relatively high loss at convergence, the performance of the jointly trained model is lower than our two-stage model on the two SAT datasets, as shown in Table 8.

Table 7: The Training Loss of End-to-End Joint Training

| Training Step | End-to-End DPO Loss | End-to-End SFT Loss |
|---|---|---|
| 100 | 0.54 | 1.64 |
| 200 | 0.60 | 1.53 |
| 300 | 0.26 | 1.44 |
| 400 | 0.43 | 1.28 |
| 500 | 0.56 | 1.14 |
| 600 | 0.13 | 1.04 |
| 700 | 0.42 | 0.98 |
| 800 | 0.29 | 0.88 |
| 900 | 0.30 | 0.86 |

Table 8: The Jointly Trained Model's Performance over Two SAT datasets w.r.t. PAR-2 Score

| Dataset | Two-Stage | Joint End-to-End |
|---|---|---|
| **Zamkeller** | 270.62 | 751.34 |
| **PRP** | 1804.46 | 2092.71 |

## J  Experiment on the Training Strategy

To provide a clearer and more comprehensive answer, we offer a detailed analysis focusing on dataset complexity and the iterative optimization behavior of our proposed method, strcmp, and its variants.

**Dataset Complexity.** The complexity of combinatorial optimization problems can be characterized by various metrics. Drawing from established literature, we focus on several key graph-based metrics—Modularity, Density, and Clustering—to provide a quantitative perspective on the structural complexity of the datasets used in our experiments. While not absolute, higher values in these metrics generally suggest a more intricate problem structure and complexity. The problem sizes are detailed in Appendix D of our manuscript. The statistics for the MILP and SAT datasets are presented below.

Table 9: Data Complexity of MILP Dataset

| Dataset | Modularity | Density | Clustering |
|---|---|---|---|
| setcover | 0.18 | 0.07 | 0.34 |
| knapsack | 0.07 | 0.04 | 0.36 |
| mis | 0.48 | 0.01 | 0.02 |
| mik | 0.11 | 0.23 | 0.30 |
| loadbalancing | 0.49 | 0.01 | 0.01 |

Table 10: Data Complexity of SAT Dataset

| Dataset | Modularity | Density | Clustering |
|---|---|---|---|
| CNP | 0.8181 | 0.0016 | 0.2012 |
| CoinsGrid | 0.9306 | 0.0005 | 0.5377 |
| Zamkeller | 0.7383 | 0.0004 | 0.5659 |
| PRP | 0.8865 | 0.0002 | 0.5429 |

**Iterative Behavior of STRCMP and its Variants.** To provide a clearer picture, we have compiled tables showing the non-smoothed, iteration-by-iteration performance on both MILP and SAT datasets. We employ an early stopping strategy where the search terminates if no performance improvement is observed within three consecutive iterations. Hence the total iteration number of different method will be varied. These results illustrate the performance trajectory (Primal-Dual Integral and PAR-2 score) of STRCMP against its SFT-only and DPO-only ablations, as shown in the following tables.

Table 11: The Performance v.s. Iteration Result w.r.t. Primal-Dual Integral between Different Methods over corlat Dataset

| Iterations | LLM4Solver | STRCMP | STRCMP (SFT Only) | STRCMP (DPO Only) | STRCMP w/o GNN |
|---|---|---|---|---|---|
| 3 | 9939.48 | 8750.84 | 6880.25 | 7159.24 | 10325.77 |
| 6 | 9860.20 | 8889.19 | 7235.31 | 6870.31 | 10435.85 |
| 9 | 9673.47 | 8836.85 | 7364.47 | 6881.48 | 10404.27 |
| 12 | 9900.06 | | | 6932.57 | |

Table 12: The Performance v.s. Iteration Result w.r.t. Primal-Dual Integral between Different Methods over mik Dataset

| Iterations | LLM4Solver | STRCMP | STRCMP (SFT Only) | STRCMP (DPO Only) | STRCMP w/o GNN |
|---|---|---|---|---|---|
| 3 | 535.04 | 478.08 | 394.25 | 451.48 | 492.96 |
| 9 | 499.60 | 455.18 | 421.44 | 434.09 | 494.24 |
| 15 | 470.48 | 450.72 | 408.55 | 431.42 | 481.97 |
| 21 | 466.45 | 438.51 | 372.12 | | |
| 27 | 471.22 | 448.31 | 380.74 | | |

Table 13: The Performance v.s. Iteration Result w.r.t. Primal-Dual Integral between Different Methods over loadbalancing Dataset

| Iterations | LLM4Solver | STRCMP | STRCMP (SFT Only) | STRCMP (DPO Only) | STRCMP w/o GNN |
|---|---|---|---|---|---|
| 3 | 1942.27 | 1881.74 | 2125.22 | 2193.62 | 1910.53 |
| 9 | 1885.31 | 1878.52 | 2065.61 | 2245.97 | 1905.49 |
| 15 | 1867.71 | 1859.14 | 2051.36 | | |

Table 14: The Performance v.s. Iteration Result w.r.t. PAR-2 Score between Different Methods over CNP Dataset

| Iteration | STRCMP | STRCMP (SFT Only) | STRCMP (DPO Only) |
|---|---|---|---|
| 3 | 49931 | 49395.1667 | 48963.5 |
| 15 | 60433.3333 | 59101.3333 | 57184 |
| . . . | . . . | . . . | . . . |
| 51 | 57108.1667 | 51935.6667 | 57688.67 |
| 57 | 55257.8333 | 52890.1667 | 57352 |
| . . . | . . . | . . . | . . . |
| 75 | | | 48779.5 |
| 78 | | | 47443 |

**Analysis and Insights.** From the iterative performance results, we draw two primary observations:

❶ SFT-only models often converge faster in terms of iteration count but may stabilize at a suboptimal performance level. This suggests that while SFT is effective at rapidly imitating strong solutions, it may overfit to the strategies present in the initial dataset, leading to premature convergence in a local optimum.

❷ DPO-inclusive models (DPO-only and SFT+DPO) demonstrate a capacity for continued improvement over a longer training horizon, often surpassing the SFT-only model's peak performance. We posit that DPO enhances the model's generalization capabilities. By learning from preference pairs rather than absolute solutions, the model develops a more nuanced understanding of what constitutes a better search trajectory, allowing it to escape local optima and discover superior solutions.

Based on these observations and the complexity analysis, we can summarize our insights on selecting a training strategy:

❶ For "easy" datasets (e.g., small-scale, sparse, solvable to optimality quickly): The SFT-only approach is often sufficient and efficient. On these problems, high-quality training data (i.e., optimal or near-optimal solutions) can be generated at a low cost, allowing SFT to quickly learn an effective policy.

❷ For "hard" datasets (e.g., large-scale, dense, complex structure, intractable): We strongly recommend a DPO-based approach (DPO-only or SFT+DPO). For these problems, obtaining optimal solutions for SFT is prohibitively expensive or impossible. However, generating preference pairs by comparing the performance of different candidate solutions is still feasible and relatively cheap. The superior generalization of DPO-trained models makes them better suited to navigating the vast and complex search spaces of these challenging instances.

❸ For medium complexity datasets, the choice is less definitive. As the results show, the interplay between SFT and DPO is intricate. Our full STRCMP model (SFT+DPO) often acts as a robust default, leveraging SFT for a strong initialization and DPO for refinement and generalization. However, we acknowledge that determining the optimal blend is a significant challenge. The task-specific training of large models has not yet fully clarified the precise relationship between SFT and DPO—when one is definitively superior, or how their data should be optimally combined. We consider this an important open problem for the field.

Table 15: The Performance v.s. Iteration Result w.r.t. PAR-2 Score between Different Methods over CoinsGrid Dataset

| Iteration | STRCMP | STRCMP (SFT Only) | STRCMP (DPO Only) |
|---|---|---|---|
| 3 | 60376 | 59444 | 59735.67 |
| 9 | 65177.67 | 65305 | 70214.83 |
| ... | ... | ... | ... |
| 30 | 58516.83 | 62026.17 | 61296.83 |
| 33 | 58091.67 | 62460.83 | 61622.17 |
| ... | ... | ... | ... |
| 57 | | 56775.17 | 53737.67 |
| 60 | | | 58212.83 |

Table 16: The Performance v.s. Iteration Result w.r.t. PAR-2 Score between Different Methods over PRP Dataset

| Iteration | STRCMP | STRCMP (SFT Only) | STRCMP (DPO Only) |
|---|---|---|---|
| 3 | 53808.17 | 53696.33 | 54977.5 |
| 9 | 54523 | 56456.67 | 54462.5 |
| ... | ... | ... | ... |
| 39 | 53015.83 | 55141.67 | 52635 |
| 42 | 50812.5 | 53459 | 52731.67 |
| ... | ... | ... | ... |
| 66 | 52701.5 | 53627 | 51030 |
| 69 | | | 50352.33 |
| ... | ... | ... | ... |
| 87 | | | 49281 |

Table 17: The Performance v.s. Iteration Result w.r.t. PAR-2 Score between Different Methods over Zamkeller Dataset

| Iteration | STRCMP | STRCMP (SFT Only) | STRCMP (DPO Only) |
|---|---|---|---|
| 3 | 45613.5 | 44988 | 45427 |
| 6 | 56487.17 | 47457.5 | 51138 |
| ... | ... | ... | ... |
| 27 | 56511.5 | 49499.17 | 44826.5 |
| 30 | 54933.67 | 50914 | 45185.67 |
| ... | ... | ... | ... |
| 51 | 51801.33 | 53002.83 | 42444.17 |
| 54 | 51900.67 | 49976.5 | 38346.17 |
| ... | ... | ... | ... |
| 72 | | | 41710 |
| 75 | | | 36365 |

## K Dependence on the Underlying LLM

Given the time and resource constraints, we focused our evaluation on the Llama2 and Qwen2 families of models, selecting representatives of varying sizes. We performed training and testing on two datasets from the MILP domain and two from the SAT domain. The results are presented below.

It is important to note that the entire Llama2 family of models failed on our code generation task, either by exceeding the context limitation of 4096 tokens or by being unable to generate syntactically correct, executable code. Similarly, the smaller Qwen2 models (0.5B and 1.5B) were also unable to produce viable code for the solvers. This underscores the complexity of the task, which requires a highly capable code-generating LLM. From the results in the tables, we can draw the following conclusions:

Table 18: The Performance of STRCMP w.r.t. Primal-Dual Integral with Varied-Size Backbone LLM over MILP Domain

| | | | STRCMP | | STRCMP **w/o GNN** | |
| --- | --- | --- | --- | --- | --- | --- |
| **Dataset** | Qwen2(0.5B) | Qwen2(1.5B) | Qwen2(7B) | Qwen2(14B) | Qwen2(7B) | Qwen2(14B) |
| mik | – | – | 269.68 | 233.56 | 292.41 | 262.74 |
| setcover | – | – | 34.02 | 34.29 | 34.06 | 34.24 |

Table 19: The Performance of STRCMP w.r.t. PAR-2 Score with Varied-Size Backbone LLM over SAT Domain

| | | | STRCMP | | STRCMP **w/o GNN** | |
| --- | --- | --- | --- | --- | --- | --- |
| **Dataset** | Qwen2(0.5B) | Qwen2(1.5B) | Qwen2(7B) | Qwen2(14B) | Qwen2(7B) | Qwen2(14B) |
| PRP | – | – | 470.63 | 414.12 | 500.00 | 455.09 |
| Zamkeller | – | – | 355.3 | 127.7 | 635.75 | 308.30 |

❶ The structural embedding from the GNN consistently and significantly contributes to the final performance. Across both MILP and SAT domains, and for both the 7B and 14B model sizes, STRCMP consistently outperforms its STRCMP w/o GNN variant. This confirms that the GNN provides a crucial inductive bias that guides the LLM toward better solutions, regardless of the LLM's scale.

❷ STRCMP 's performance is robust, provided a sufficiently capable LLM backbone is used. While our framework's effectiveness is contingent on the LLM's fundamental ability to comprehend the task and generate valid code, the performance is stable across capable models of different sizes.

❸ STRCMP 's performance is sensitive to the size of the LLM backbone, but this effect is problem-dependent. The results reveal a nuanced relationship between model scale and performance. For the more complex mik instances, scaling the LLM from 7B to 14B parameters yields a notable performance gain. However, for the setcover instances, the 7B and 14B models perform almost identically. This suggests that while larger models can unlock better performance on certain complex problem structures, a more moderately-sized model may be sufficient for others.

## L Prompt and Corresponding Responses

---

**Prompt of data curation for SAT domain (adapted from AutoSAT [29])**

```
# Description of the role duty
You are a SAT solver researcher trying to write the bump_var to help
↪  SAT solver escape from local optimum.
# Description of the task goal
Your goal is to improve bump_var for the SAT solver, after reading
↪  and understanding the <key code> of SAT solver below.

# Description of the example codes
As an example, the format of bump_var can be:
'''
// start
void Solver::bump_var(int var, double coeff) {
    ...
}
// end
'''

To replace the original code:

'''
// start
void Solver::bump_var(int var, double coeff) {
    if ((activity[var] += var_inc * coeff) > 1e100) {
    // Update score and prevent float overflow
        for (int i = 1; i <= vars; i++) activity[i] *= 1e-100;
        var_inc *= 1e-100;}
    if (vsids.inHeap(var)) vsids.update(var);
    // update heap
}
// end
'''

# Description of the tips to the task
Tips:
1) Your bump_var code must start with '''// start''' and end with
↪  '''// end'''
2) Feel free to unleash your creativity and provide your improved and
↪  correct code. (Pay attention to correct namespace.)
3) Your bump_var code must be different from original codes.
4) You can only modify the current bump_var and use the existing
↪  external variables (e.g. variables that defined in Solver).
5) You cannot add new external variables outside the bump_var.
6) The SAT problems that the SAT solver needs to solve have the
↪  structural characteristics.

# The whole code of target solver
<key code> of SAT solver is:
"""
// Omit the key code of SAT solver here for space limitations.
"""

Take a deep breath and think it carefully. Then respond in correct
↪  format!
```

---

### Collected Response to above Prompt

```cpp
// start
void Solver::bump_var(int var, double coeff) {
    // Update the activity score with a dynamic coefficient based on
    ↪  LBD
    double dynamic_coeff = coeff * (1.0 + (clause_DB.back().lbd /
    ↪  10.0));
    if ((activity[var] += var_inc * dynamic_coeff) > 1e100) {
    ↪  // Update score and prevent float overflow
        for (int i = 1; i <= vars; i++) activity[i] *= 1e-100;
        var_inc *= 1e-100;
    }
    if (vsids.inHeap(var)) vsids.update(var);
    // update heap
    // Implement a decay factor for activity scores based on the
    ↪  number of conflicts
    static int conflict_count = 0;
    conflict_count++;
    if (conflict_count % 1000 == 0) {
        for (int i = 1; i <= vars; i++) activity[i] *= 0.95;
        // Decay activity scores
    }
}
// end
```

**Prompt of data curation for MILP domain (adapted from previous work [15, 30])**

# Description of the task background
1. Introduction to MILP: Mixed Integer Linear Programming (MILP) is a
↪ mathematical optimization or decision making methodology for
↪ mathematical planning problems in which the objective function
↪ and constraints are linear and some of the decision variables are
↪ restricted to integers.
2. MILP Definition: A mixed integer linear programming (MILP) problem
↪ has the following elements:
    2.1) A linear objective function fx, where f is a column vector
    ↪ of constants and x is a column vector of unknowns.
    2.2) Boundary and linear constraints, but no nonlinear
    ↪ constraints.
    2.3) Restrictions on certain components of x so that they must
    ↪ have integer values.
3. Cutting Planes Definition: cutting planes are additional linear
↪ inequality constraints added to the MILP problem. These
↪ inequalities attempt to restrict the feasible domain of the LP
↪ relaxation so that its solution is closer to an integer.
4. Cutting Plane Algorithm: The Cutting Plane method solves the MILP
↪ problem by linearly relaxing the integer problem into a
↪ non-integer linear problem and solving it. The theory of linear
↪ programming states that under mild assumptions (if there exists
↪ an optimal solution to the linear programming and the feasible
↪ domain does not contain a line), there always exists an extremum
↪ or vertex that is optimal. Test whether the optimal solution
↪ obtained is an integer solution. If not, then there must exist a
↪ linear inequality separating the optimal point from the convex
↪ envelope of the true feasible set. Finding such an inequality is
↪ the separation problem, and such an inequality is the cutting
↪ plane. The cutting plane can be added to the linear program being
↪ relaxed such that the current abelian solution is no longer
↪ feasible for the relaxation. The process is repeated until an
↪ optimal integer solution is found.
5. Instructions: You need to understand the above, especially about
↪ the cut-plane algorithm, the next tasks will be closely related
↪ to the cut-plane algorithm.

# Description of task details
1. Sample code for a cut plane selection function:

```
[start]
import numpy as np
import ... as ...
def fun_name(cuts_features, sel_cuts_num):
...
return idxes
[end]
```

The parameters of the function are the characteristics of all cuts
↪ and the number of cuts to be selected. After completing the
↪ selection, it returns an idx list of all selected cutting planes.
↪ In addition, each function is independent, each function needs to
↪ import the corresponding package such as numpy... and each
↪ function uses [start] and [end] as the start and end identifiers.

**(Continued) Prompt of data curation for MILP solver**

2. Characterization: Pay attention to the cutting plane features in
↪ the function input.The following are the specific contents
↪ contained in these features:

```
cut_feature = [
obj_parall,                    # Target Parallelism
eff,                           # Cut-plane validity
support,
integral_support,
normalized_violation,
mean_coeff_cut,
max_coeff_cut,
min_coeff_cut,
std_coeff_cut,
mean_coeff_obj,
max_coeff_obj,
min_coeff_obj,
std_coeff_obj
]
```

3. No comments or annotations are required, only the code needs to be
↪ provided.

**Collected Response to above Prompt**

```
[start]
import numpy as np
def hybrid_diversity_selector(cuts_features, sel_cuts_num):
    features = np.array(cuts_features)
    if len(features) <= sel_cuts_num:
        return list(range(len(features)))

    eff = features[:, 1]
    viol = features[:, 4]
    supp = features[:, 2] + features[:, 3]
    coeff_std = features[:, 8] / (features[:, 5] + 1e-10)
    quality = np.sqrt(eff * viol) * np.log1p(supp)
    q_thresh = np.percentile(quality, 70)
    var_thresh = np.percentile(coeff_std, 30)
    mask = (quality >= q_thresh) & (coeff_std <= var_thresh)
    candidates = np.where(mask)[0]

    if len(candidates) <= sel_cuts_num:
        return candidates.tolist()

    active_features = features[candidates]
    diversity = np.zeros(len(candidates))
    for i in range(len(candidates)):
        coeff_diff = np.sum(np.abs(active_features[:,5:9] -
        ↪ active_features[i,5:9]), axis=1)
        obj_diff = np.abs(active_features[:,0] -
        ↪ active_features[i,0])
        diversity[i] = np.mean(coeff_diff * obj_diff)
    scores = quality[candidates] * diversity
    return candidates[np.argsort(-scores)[:sel_cuts_num]].tolist()
[end]
```

