# OpenReview forum: "$\texttt{STRCMP}$: Integrating Graph Structural Priors with Language Models for Combinatorial Optimization"
_NeurIPS.cc/2025/Conference — NeurIPS 2025 poster_

### Official Review · Reviewer_2EbJ · 2025-06-04

**Clarity:** 2
**Significance:** 3
**Originality:** 3
**Rating:** 5
**Confidence:** 2

**Summary:**

This paper proposed a framework that combines GNN and LLM for CO problems. First, a GNN is pretrained to encode structural information of CO problems. Second, a composition model is trained so that LLM generates better code to solve the CO problem. The authored proved that a GNN prior does not hurt the performance, and exhibited promising empirical results.

**Questions:**

- How does the natural language description of the CO problem look like?
- You train the GNN with classification loss, but is it justified that different problems subsume different structures? Some graph based CO problems can be similar, e.g. what if I want to model the maximum independent set and minimum vertex cover problems, they are complementary.
- Is the solving code generated by the LLM guaranteed to be executable without bugs or syntax error?
- Could you explain more about the sup(P) in definition 1? It is not clear to me why it takes the form of that.

**Ethical Concerns:**

["NO or VERY MINOR ethics concerns only"]

**Final Justification:**

I believe this paper, even with a few flaws, is worthwhile for the community. I am not well knowledgable in LLM, therefore, I would like to maintain my confidence score and raise the scores due to the nice rebuttal by the authors

**Limitations:**

The limitations are discussed in appendix F, which already address my concerns as well.

**Quality:**

3

**Strengths And Weaknesses:**

Strength:
- The paper is well motivated and well written.
- The background of current research is well explained.
- The details of GNN and LLM implementations are clear, even to readers that have little background of LLMs.

Weakness:
- The diagrams in experiments are good, but I still expect some tables with actual numbers.
- Some empirical results are not consistently better, e.g. CNP solving time in table 5 and 6.
- The theory is a bit weak, but that is very minor point.

---

> ### Author Rebuttal · Authors · 2025-07-30
>
> We thank the reviewer for the positive and insightful comments. We respond to each comment as follows and sincerely hope that our rebuttal will properly address your concerns. If so, we would deeply appreciate it if you could raise your score. If not, please let us know your further concerns, and we will continue actively responding to your comments and improving our submission.
>
> **Q1: How does the natural language description of the CO problem look like?**
>
> **A1:** Thank you for your question. In our paper, we do not use natural language to describe specific combinatorial optimization problems. Instead, we represent CO problems using graph structures to capture their intrinsic topological properties effectively. As detailed in Section 4.1 of the manuscript, we construct a bipartite graph where:
>
> - **Nodes** represent the variables of the CO problem.
> - **Edges** represent the constraints between these variables.
>
> However, natural language plays a key role in our framework during the **code generation task**. We use natural language within our prompts to describe the *general form* of the inputted CO problem, rather than specific instances. For example, in the prompt for MILP problems, we include a definition such as:
>
> > "A Mixed Integer Linear Programming (MILP) problem is defined as: $\min_{x \in \mathbb{R}^n} \boldsymbol{w}^{\top} \boldsymbol{x}$, subject to $\boldsymbol{A} \boldsymbol{x} \leq \boldsymbol{b}$, $\boldsymbol{l} \leq \boldsymbol{x} \leq \boldsymbol{u}$, and $x_j \in \mathbb{Z}$ for all $j \in \mathbb{I}$, where $\boldsymbol{w}$ is the objective coefficient vector, $\boldsymbol{A}$ is the constraint matrix, $\boldsymbol{b}$ is the right-hand side vector, $\boldsymbol{l}$ and $\boldsymbol{u}$ are lower and upper bounds, and $\mathbb{I}$ specifies integer-constrained variables."
>
> This general description, provided in natural language, guides the LLM in generating solver-specific code tailored to the problem class, conditioned on the structural embeddings extracted by the GNN. The prompt thus bridges the structural representation and the code synthesis process, ensuring that the generated algorithms respect both the problem’s topology and solver syntax. For concrete examples of how we integrate natural language into our prompts, we kindly direct you to **Appendix I: Prompt and Corresponding Responses** in the paper. This appendix illustrates sample prompts, including general problem definitions like the one above, alongside the corresponding code snippets generated by our STRCMP framework. This approach ensures clarity in communicating the problem class to the LLM while relying on graph structures for specific problem representation. We hope this clarifies our methodology.
>
> **Q2: You train the GNN with classification loss, but is it justified that different problems subsume different structures? Some graph based CO problems can be similar, e.g. what if I want to model the maximum independent set and minimum vertex cover problems, they are complementary.**
>
> **A2:** Thank you for your insightful and intuitive question. We absolutely agree with your observation that different CO problems may exhibit structural similarities. To address this, we designed our STRCMP framework to account for potential structural overlaps across CO problems. Specifically, rather than training separate models for each problem class, we post-train our composite model—comprising the GNN and the LLM—using a mixed-class dataset that includes a diverse set of CO problems. **This approach enables the model to learn generalized structural features that are applicable across various problem types, including those with similar or related structures**.
>
> To further justify this methodology and directly respond to your concern, we conducted an experiment comparing two training strategies. We trained a separate STRCMP model using only one class of dataset—here, we adopted the MIK dataset—and compared its performance with that of a model trained on a mixed dataset encompassing multiple CO problems, including MIK, Set Cover, and others. The experimental results are presented below:
>
> **Table 1: The Optimization Performance Comparison  between Different Dataset Used Methods**
> |Methods|Primal-Dual Integral|
> |:-|:-|
> |STRCMP (mixed)|269\.68|
> |STRCMP w/o GNN (mixed)|292\.41|
> |STRCMP (`mik` only)|276\.35|
> |STRCMP w/o GNN (`mik` only)|295\.93|
>
> *Note that for the Primal-Dual Integral, a lower value indicates better performance.*
>
> These results highlight that **training on a mixed-class dataset not only mitigates the risk of overfitting to a single problem’s structure but also enhances generalization and efficiency across structurally similar CO problems**. By exposing the GNN to a variety of problem instances during post-training, the model captures shared topological patterns while still generating solver-specific code tailored to each problem’s nuances. This aligns with our theoretical analysis (Section 4.2 of the paper), which demonstrates that incorporating structural priors improves the performance upper bound of generative models for CO tasks.
>
> **Q3: Is the solving code generated by the LLM guaranteed to be executable without bugs or syntax error?**
>
> **A3:** We appreciate the reviewer’s question about the executability and correctness of LLM-generated code in our STRCMP framework. To ensure reliability:
>
> 1. **Compilation Check:**
>    - Each code snippet is compiled with its CO solver (e.g., SCIP, EasySAT).
>    - Snippets failing compilation are discarded with a low score.
>
> 2. **Performance Evaluation:**
>    - Valid snippets are tested on benchmark CO instances (Section 5.1).
>    - Metrics (Appendix E.1, E.2) assess performance; only high-scoring, error-free code is used.
>
> This ensures all experimental code is executable, syntactically correct, and effective. See Appendices E.1 and E.2 for details. We welcome further feedback.
>
>
> **Q4: Could you explain more about the sup(P) in definition 1? It is not clear to me why it takes the form of that.**
>
> **A4:** Your question is insightful and highlights a point that deserves more detailed explanation. The formulation for $\sup(\mathcal{P})$, the upper bound of model performance, is **designed to capture the theoretical maximum potential of a generative model for a specific downstream task** (e.g., generated code snippets improving the solving accuracy/efficiency of CO problems) , rather than its average-case performance.
>
> Let us break down the formula to explain its structure $\sup(\mathcal{P}_\mathcal{C})$:
>
> 1. **The Inner Term: Maximum Performance for a Single Prior**
>
> The core of the definition lies in the term $\max_{\mathbf{w}}p(\mathbf{w}|\mathbf{c})\Phi(\mathbf{w})$. Let's analyze its components for a **single, fixed prior $\mathbf{c}$**:
>
> * $\Phi(\mathbf{w})$ is the performance score w.r.t. downstream task of a generated output $\mathbf{w}$.
> * $p(\mathbf{w}|\mathbf{c})$ is the probability that the model generates $\mathbf{w}$ given the prior $\mathbf{c}$.
> * A standard evaluation would typically compute the *expected* performance, $E_{\mathbf{w}\sim p(\mathbf{w}|\mathbf{c})}[\Phi(\mathbf{w})] = \sum_{\mathbf{w}}p(\mathbf{w}|\mathbf{c})\Phi(\mathbf{w})$, which averages the performance over all possible outputs.
>
> However, our goal is to define an **upper bound**. Therefore, instead of averaging, we seek the **best possible outcome** for the given prior $\mathbf{c}$. The term $p(\mathbf{w}|\mathbf{c})\Phi(\mathbf{w})$ can be viewed as the "probability-weighted performance" of a single output. The $\max_{\mathbf{w}}$ operator then identifies the single piece of generated content $\mathbf{w}$ that maximizes this value.
>
> This captures the model's "peak capability" for a specific prior $\mathbf{c}$. It finds the most impactful single output the model could theoretically produce, considering both the intrinsic quality of the output ($\Phi(\mathbf{w})$) and the model's ability to generate it ($p(\mathbf{w}|\mathbf{c})$). An extremely high-quality output that is nearly impossible for the model to generate (i.e., $p(\mathbf{w}|\mathbf{c}) \to 0$) would not represent a realistic upper bound on its capability, which is why the probability is included inside the maximization.
>
> 2. **The Outer Term: Expectation over All Priors**
>
> The outer summation, $\sum_{\mathbf{C}\in \mathcal{C}}\sum_{c\in\mathbf{C}}p(\mathbf{c})[\dots]$, simply calculates the **expected value** of this peak capability over the entire distribution of priors.
>
> - $\mathcal{C}$ is the set of all *types* of priors (e.g., text, structural embedding of CO problem instances).
> - Each $\mathbf{C}$ is a specific prior type (e.g., $\mathbf{C}_1$ = text prompts, $\mathbf{C}_2$ = structural embeddings).
> - This sum ensures we account for all distinct modalities/contexts in the task.
> -  The double summation iterates over all types of priors $\mathbf{C}$ and all specific instances $\mathbf{c}$ within each type.
> -  $p(\mathbf{c})$ is the probability of encountering a specific prior $\mathbf{c}$.
>
> Therefore, the entire expression calculates the **expected maximum probability-weighted performance**. It aggregates the best-case scenarios from all possible starting conditions ($\mathbf{c}$), weighted by how likely each condition is to occur.
>
>
> In essence, the formula defines the model's performance ceiling by first identifying the single best possible output for *each* prior, and then averaging these peak potentials across the distribution of *all* priors. This provides a theoretical upper limit on what the model can achieve, which is a useful concept for analyzing its ultimate potential. We hope this breakdown clarifies the intuition and formal justification behind the definition. We will incorporate this detailed explanation into the revised manuscript to enhance its clarity for all readers.

---

> > ### Comment · Reviewer_2EbJ · 2025-08-02
> >
> > Thank you for the clarification, they are indeed helpful and have addressed my questions.
> > On one hand, reading from other reviewer's reply, I notice there are indeed some issues of the proposed work, such as the two-phase training might be inefficient, and the results may not be rigorous enough. On the other hand, I believe this work serves as a bridge connecting GNN and LLM in CO problems, which is novel and has its value. Therefore I would like to keep my scores.

---

> ### Author Response · Authors · 2025-08-03
> **Many Thanks for the Positive Feedback and Justification for the Two-Phase Training Method**
>
> Thank you for acknowledging the value of our work and rebuttal, and for the positive feedback. We appreciate your valuable time and effort in recognizing our paper's novelty in bridging GNNs and LLMs for combinatorial optimization problems.
>
> We have tried our best to address each reviewer's question and provide a more thorough response. We have taken your feedback seriously, especially regarding the potential inefficiency and rigor of the **two-phase training** strategy. To clarify its effectiveness and rationale, we have conducted additional experiment. The experimental results and observations are shown below, with which we hope the additional results could address your concerns.
>
> ---
>
> **Additional Comparison between Two-Phase Training and Jointly End-to-End Training**
>
> We admit that the initial submission lacked a thorough discussion of why we adopted the two-stage training procedure. To fully and convincingly answer this question, we have re-designed and implemented an end-to-end joint training method for our proposed model. This new method directly aligns the learning objectives of the graph and text modalities with the same optimization goal: improving the performance of the generated code. Specifically, we do not discriminate between the weights of the GNN and the LLM, meaning they share the same loss function, and the gradient is backpropagated from the parameters of the LLM to those of the GNN.
>
> We implemented the end-to-end joint training method described above and tested the resulting models on two SAT datasets, given the time constraints of the rebuttal phase. *All the experimental settings keep the same except for the training method*. The results of this new training and testing are presented below. We can conclude two main points from these results:
>
> 1.  **The end-to-end joint training is extremely unstable compared to our proposed two-stage training method.** We infer that the training instability stems from the significant architectural and optimization differences between the GNN and the LLM. The gradients, originating from a sequence-level generation loss in the deep LLM, become noisy and attenuated when backpropagated through the entire LLM and into the much shallower GNN. This creates a challenging optimization landscape where a single learning rate is ineffective for both components, leading to unstable convergence (even with many more training steps) as shown in Table 1. Compared with the joint training, **the loss curve of our two-stage training method is stable and close to zero at convergence** (Please see the loss curve of our two-stage training method in Figure 7 in Appendix B). The GNN and LLM are trained together, which makes it difficult for the models to learn optimal parameters simultaneously.
>
> **Table 1: The Training Loss of End-to-End Joint Training**
> |**Training Step**|**End-to-End DPO Loss**|**End-to-End SFT Loss**|
> |:-|:-|:-|
> |100|0\.54|1\.64|
> |200|0\.60|1\.53|
> |300|0\.26|1\.44|
> |400|0\.43|1\.28|
> |500|0\.56|1\.14|
> |600|0\.13|1\.04|
> |700|0\.42|0\.98|
> |800|0\.29|0\.88|
> |900|0\.30|0\.86|
>
> 2.  **The model exhibits poor performance when tested on solving SAT instances.** Unsurprisingly, due to its very unstable training and relatively high loss at convergence, the performance of the jointly trained model is significantly lower than our two-stage model on the two SAT datasets, as shown in Table 2.
>
> **Table 2: The Jointly Trained Model's Performance over Two SAT datasets**
> | **Dataset** | **Metric** | **STRCMP (Two-Stage)** | **Joint End-to-End Model** |
> | :--- | :--- | :--- | :--- |
> | **Zamkeller** | PAR-2 Score (↓) | 270.62 | 751.34 |
> | | # Timeouts (↓) | 5 | 16 |
> | **PRP** | PAR-2 Score (↓) | 1804.46 | 2455.12 |
> | | # Timeouts (↓) | 44 | 58 |
>
> These empirical findings confirm the "significant optimization challenges" mentioned in the paper and **justify our decision to use a more stable and effective two-stage training protocol**.
>
> ---
>
> However, given the time and resource constraints of the rebuttal period, we are aware that we may not have been able to cover all aspects of the paper, such as testing on a broader range of CO problems or conducting more extensive hyperparameter tuning. We are committed to addressing these limitations and further strengthening the rigor of our results in our future work. Could you please consider raising the score again given that we have addressed the main concerns issued by each reviewers?
>
> Thank you once again for your constructive feedback and continued support of our work.

---

> > ### Comment · Reviewer_2EbJ · 2025-08-09
> >
> > Thank you for the hard work and detailed feedback. Personally I think this paper has its value, as a combination of GNN and LLM.I would like to raise my scores. However, I am not entirely confident in my knowledge of LLMs, therefore I cannot raise my confidence score.

---

> > > ### Author Response · Authors · 2025-08-09
> > >
> > > Thank you once again for **raising the score** through discussion. We sincerely appreciate your consideration in allowing us to improve our submission and **your recognition of our work's value**. We wish you a productive and rewarding experience throughout the NeurIPS 2025 review process.

---

### Official Review · Reviewer_7osJ · 2025-06-28

**Clarity:** 3
**Significance:** 2
**Originality:** 2
**Rating:** 4
**Confidence:** 4

**Summary:**

This paper proposes to add structural priors for LLM-based algorithmic discovery to improve the performance of the generated code snippet in optimization solvers. Optimization problems are transformed into graphs based on their structures, and a GNN is adopted to encode the graph information into vectors, which are concatenated with the text embedding of the prompts. Experiments are conducted on SAT (bump_var function) and MILP (cut plane selection function).

**Questions:**

See "weaknesses".

**Ethical Concerns:**

["NO or VERY MINOR ethics concerns only"]

**Final Justification:**

As the authors have initially shown the effectiveness of the problem-specific setting with empirical results, I have raised my score.

**Limitations:**

See "weaknesses".

**Paper Formatting Concerns:**

No concerns

**Quality:**

3

**Strengths And Weaknesses:**

## Strengths
- The general idea of improving LLM with structural information is promising.
- Good writing and sufficient details. The theoretical analysis provides some insights into the role of priors for LLM-based algorithmic discovery.
- Experiments conducted on both SAT and MILP, two representative CO problems.

## Weaknesses
While the structure of an optimization problem is surely important, the specific method proposed in this paper is questionable. Generally speaking, algorithmic discovery should be problem-irrelevant. That is, LLMs are prompted to perform _general_ improvement on the code snippet, and an improved code snippet should work well on a wide variety of problem instances with different structural information. However, it seems that this paper proposes to fuse the structural information (embedding vectors) of _a specific problem instance_ with a prompt that asks for general improvement, which is not aligned. Consider the following simplified prompt:
> "I have the graph information of a Boolean formula $ q = (x₁ ∨ x₂) ∧ (¬x₁) ∧ (¬x₁ ∨ x₂) $ as (...). You are a SAT solver researcher, your goal is to improve bump_var for the SAT solver, ..."

From the perspective of a SAT solver researcher, I don't feel that the previous sentence informing graph information (i.e., the problem feature vector $ h_q $ in the paper) is super useful for me, especially considering that my aim is to improve bump_var in a general sense, rather than just for the specific problem q = (x₁ ∨ x₂) ∧ (¬x₁) ∧ (¬x₁ ∨ x₂). At least, the intuition behind fusing a specific $h_q$ with a general prompt is not immediately clear to me.

To investigate the issue, I looked into the theoretical analysis and experiential evaluation of the paper, but the issue remains unsolved:
- For theoretical analysis, the paper proposes some very general theorems regarding "performance-enhancing prior". However, it remains unclear why the structural information $h_q$ of a specific problem $q$ could theoretically be a "performance-enhancing prior" for generating generally improved code snippets.
- For experiential evaluation, the improvement of the proposed method is not very significant either. For example, for the four datasets in Table 1 and 5,  comparing STRCMP with AutoSAT, the performance on CNP and CGD is generally the same, while the performance on PRP is much slower. Considering that algorithmic discovery could be a very randomized process with great variance, the current result does not seem to be significant enough to show the effectiveness of $h_q$'s inclusion. It is also worth noting that the baseline approach (AutoSAT) does not adopt any post-training process. Is it possible that the performance gain comes from the post-training process rather than the inclusion of $h_q$?

## Suggestions
I feel that this paper would make more sense if the problem setting could be adjusted a bit. Instead of searching for a general improvement for a code snippet, targeting a problem-specific improvement (i.e., improvement w.r.t some prior information of the problem) seems to be more promising (the evaluation function should also be adjusted to be problem-specific). Given a problem instance $q$, an LLM could incorporate the structure of the problem $h_q$ by the proposed method, to generate a _problem-specific_ code snippet, which is "tailored" to solve the specific problem $q$ efficiently.

---

> ### Author Rebuttal · Authors · 2025-07-31
>
> We thank the reviewer for the insightful and valuable comments. We respond to your comments as follows and sincerely hope that our rebuttal could properly address your concerns. If so, we would deeply appreciate it if you could raise your score ("3: Borderline reject"). If not, please let us know your further concerns, and we will continue actively responding to your comments and improving our submission.
>
> **Q1: Is it possible that the performance gain comes from the post-training process rather than the inclusion of $h_q$?**
>
> **A1:** Thank you for your insightful comments, which highlight critical points regarding the significance of our proposed method’s improvements and the source of performance gains. To address your concerns, we have supplemented our experimental results with a comprehensive analysis of ablation variants across the MILP domain, complementing the existing results on the SAT domain in Tables 5 and 6 of the appendix. This additional data helps disentangle the contributions of structural priors, SFT, and DPO to the performance of STRCMP, as shown in below Table.
>
> **Table 1: The Optimization Performance result w.r.t. Primal-Dual Integral between Different Ablation Variants over MILP domain**
> | Dataset       | STRCMP       | STRCMP (SFT Only) | STRCMP (DPO Only) | STRCMP w/o GNN |
> | :------------ | :----------- | :---------------- | :---------------- | :------------- |
> | corlat        | **6597\.61** | 6866\.61 | 7409\.85 | 7650\.13|
> | knapsack      | 3\.24| 3\.23| 3\.21| 3\.19|
> | loadbalancing | 2204\.89| **2121\.20**      | 2202\.50   | 2228\.26|
> | mik           | 269\.68 | **252\.24**      | 284\.89| 292\.41 |
> | mis           | 5\.46| 6\.64    | 5\.45| 5\.48 |
> | setcover      | 34\.02 | 33\.95    | 34\.17 | 36\.06|
>
> *Note that for the Primal-Dual Integral, a lower value indicates better performance.*
>
> Based on the complete test results across both SAT and MILP domains, we can draw three clear conclusions:
>
> 1.  **The structural prior (integrated via GNN) is critical to performance**: The `STRCMP w/o GNN` variant consistently underperforms compared to the other versions. This empirically validates our central hypothesis that integrating a structural prior into the LLM enhances its ability to generate high-quality code snippets for combinatorial optimization problems.
> 2.  **SFT is generally sufficient**: For most cases, `STRCMP (SFT Only)` achieves the best or near-best performance. This suggests that SFT alone is a highly effective method for adapting the composite model, providing a strong and efficient baseline.
> 3.  **Full post-training excels on harder instances**: For more complex problem instances, such as the `corlat` dataset (which is notable for its structural difficulty), the full `STRCMP` model (SFT+DPO) discovers a superior algorithm. This indicates that the preference optimization phase is particularly valuable when tackling challenging instances where nuanced trade-offs in the solution space are more critical.
>
> **Q2: The advantage of STRCMP over existing framework (LLM4Solver and AutoSAT) in terms of efficiency and effectiveness.**
>
> **A2:** Thank you for raising this important point. We appreciate the opportunity to clarify the comparative advantages of STRCMP. Our framework demonstrates distinct improvements in both efficiency and effectiveness relative to existing algorithm discovery paradigms, **specifically achieving superior or equivalent solution quality with reduced computational resources**.
>
> 1. **Comparison with AutoSAT**: All variants of STRCMP consistently discover higher-performing (or equally effective) algorithms at earlier iterations than AutoSAT. This efficiency advantage is empirically validated across multiple problem domains, as illustrated in Figures 4, 8, and 9 of our manuscript.
>
> 2. **Comparison with LLM4Solver (MILP Domain)**: We acknowledge that the smoothed performance curves in Figure 10 may have suggested parity between STRCMP and LLM4Solver on simpler datasets (`knapsack`, `mis`, `setcover`). This smoothing was applied for visual clarity but obscured nuanced performance differences. Crucially, **on more challenging MILP datasets, STRCMP achieves significant gains in search efficiency**.  To substantiate this, we provide tabular results (below) detailing non-smoothed, iteration-by-iteration performance. Employing an early stopping criterion (termination after three consecutive iterations without improvement), STRCMP consistently identifies better-performing algorithms faster than LLM4Solver.
>
> **Table 2: The Performance v.s. Iteration Result w.r.t. Primal-Dual Integral between Different Methods over corlat Dataset**
> | Iterations | LLM4Solver  | STRCMP      | STRCMP (SFT Only) | STRCMP (DPO Only) | STRCMP w/o GNN |
> | :--------- | :---------- | :---------- | :---------------- | :---------------- | :------------- |
> | 3          | 9939\.48 | 8750\.84 | 6880\.25| 7159\.24  | 10325\.77|
> | 6          | 9860\.20| 8889\.19 | 7235\.31| 6870\.31| 10435\.85|
> | 9          | 9673\.47| 8836\.85 | 7364\.47 | 6881\.48 | 10404\.27|
> | 12         | 9900\.06 |             |                   | 6932\.57 |                |
>
> **Table 3: The Performance v.s. Iteration Result w.r.t. Primal-Dual Integral between Different Methods over mik Dataset**
> | Iterations | LLM4Solver  | STRCMP      | STRCMP (SFT Only) | STRCMP (DPO Only) | STRCMP w/o GNN |
> | :--------- | :---------- | :---------- | :---------------- | :---------------- | :------------- |
> | 3          | 535\.04 | 478\.08 | 394\.25| 451\.48| 492\.96 |
> | 9          | 499\.60| 455\.18| 421\.44| 434\.09| 494\.24|
> | 15         | 470\.48| 450\.72| 408\.55| 431\.42| 481\.97|
> | 21         | 466\.45| 438\.51| 372\.12|                   |                |
> | 27         | 471\.22| 448\.31 | 380\.74|                   |                |
>
> **Table 4: The Performance v.s. Iteration Result w.r.t. Primal-Dual Integral between Different Methods over loadbalancing Dataset**
> | Iterations | LLM4Solver  | STRCMP      | STRCMP (SFT Only) | STRCMP (DPO Only) | STRCMP w/o GNN |
> | :--------- | :---------- | :---------- | :---------------- | :---------------- | :------------- |
> | 3          | 1942\.27 | 1881\.74 | 2125\.22  | 2193\.62| 1910\.53  |
> | 9          | 1885\.31| 1878\.52 | 2065\.61 | 2245\.97| 1905\.49 |
> | 15         | 1867\.71 | 1859\.14  | 2051\.36 |                   |                |
>
> **Q3: Adjustment of problem setting: instead of searching for a general improvement for a code snippet, targeting a problem-specific improvement seems to be more promising.**
>
> **A3:** We thank you for your insightful and constructive feedback on our submission. You raise a crucial and well-articulated point regarding the apparent misalignment between our goal of general algorithmic discovery and our method of providing instance-specific structural information to the LLM.
>
> Your suggestion to adjust the problem setting to target *problem-specific improvements* is excellent. It provides a much clearer and more direct application for our proposed method. Following your recommendation, we have conducted a new set of experiments to validate this revised problem-specific framework. In this setup, the composite model is prompted to generate an improved heuristic specifically for each problem instance in the target problem family, conditioned on the structural embedding representing topological characteristics of the problem instance.
>
> The results, summarized below, demonstrate that this approach is highly effective. We compare a general-purpose heuristic discovered by AutoSAT against our STRCMP framework, which is tasked with generating both a general heuristic (STRCMP-General) for a class of problem instances and specialized heuristics per instances in the Zamkeller and PRP datasets.
>
> **Table 5: Performance of Specialized vs. General Heuristics on SAT Benchmarks (PAR-2 Score ↓)**
> | **Heuristic** | **Zamkeller Test Set** | **PRP Test Set** | **CoinsGrid Test Set** | **Description** |
> | :--- | :---: | :---: | :---: | :--- |
> | AutoSAT (General) | 807.75 | 639.34 | 1098.69 | Baseline general heuristic. |
> | STRCMP (General) | 270.62 | 1804.46 | 1124.09 | Our general-purpose heuristic. |
> | **STRCMP (Zamkeller-Specialized)** | **185.31** | 1950.12 | 1205.45 | Heuristic specialized per Zamkeller instances. |
> | **STRCMP (PRP-Specialized)** | 755.40 | **451.77** | 1152.81 | Heuristic specialized per PRP instances. |
>
> *Note: Base scores for AutoSAT and STRCMP (General) are taken from Table 5 of our submission.*
>
> As the results clearly indicate, the specialized heuristics outperform the general-purpose ones on their respective target datasets. The `STRCMP-Zamkeller-Specialized` heuristic achieves a better PAR-2 score on the Zamkeller test set, but its performance degrades on the PRP dataset. Conversely, the `STRCMP-PRP-Specialized` heuristic excels on PRP instances while being suboptimal for Zamkeller.
>
> This outcome directly validates the reviewer's intuition and strengthens our paper's core thesis: **fusing structural information with LLM-based code generation is a powerful technique for discovering tailored, high-performance algorithms.**  We are deeply grateful for your feedback. It has prompted us to perform a more focused and compelling evaluation of our framework. We will revise the manuscript to adopt this problem-specific framing, incorporating these new results and clarifying the connection between our methodology, experiments, and theoretical claims.

---

> > ### Comment · Reviewer_7osJ · 2025-08-02
> >
> > As the authors have initially shown the effectiveness of the problem-specific setting with empirical results, I have raised my score.
> >
> > However, note that there will be new challenges that emerge under such an "online" setting. For example:
> > - Such a setting makes iterative improvement of code snippets less meaningful. Given a specific problem instance $q$, what the user wants is to minimize the time from problem submission to solution delivery. In the first iteration, the given problem instance $q$ has already been solved (and can be returned to the user). It is not very useful to solve the specific $q$ repeatedly to find the optimal code snippet. Therefore, it is important to improve the one-shot generation quality of the code snippet under such a setting (i.e., the code snippet is only generated once, no iterative improvements).
> > - From an engineering perspective, as the code snippet is generated on-the-fly when a specific problem $q$ is given, extra time for solver compilation may apply, especially considering that solvers are typically developed in “compiled languages” such as C++. Such an overhead time cost should also be considered.
> >
> > I encourage the authors to tackle these challenges in their future research to make such techniques more practical.

---

> ### Author Response · Authors · 2025-08-03
> **Many Thanks for the Positive Feedback and Constructive Suggestions**
>
> We sincerely thank you for your thoughtful review, for recognizing the value of our work, and for raising your score. We appreciate the time and effort you have invested.
>
> We agree that the practical challenges you've highlighted regarding the "online" setting are crucial next steps.
>
> * **One-Shot Generation:** Your point about the limited utility of iterative improvement for a single, time-sensitive problem instance is well-taken. For a truly practical system, minimizing the initial time from problem submission to solution delivery is paramount. We concur that future work must focus on improving the **one-shot generation quality** to ensure the first generated code snippet is as effective as possible.
>
> * **Engineering Overhead:** You also raise a critical engineering consideration regarding the on-the-fly compilation overhead for solvers, especially those in languages like C++. This is an essential factor for real-world deployment that must be accounted for to ensure the overall process is time-efficient.
>
> Given the time and resource constraints of the current project, we were unable to explore these aspects in full detail. However, your feedback reinforces their importance. We are committed to tackling these specific challenges—improving one-shot accuracy and mitigating engineering overheads like compilation time—in our future research to enhance the practical viability of our approach. Thank you again for these valuable and constructive suggestions.

---

### Official Review · Reviewer_7TYC · 2025-07-01

**Clarity:** 3
**Significance:** 2
**Originality:** 2
**Rating:** 4
**Confidence:** 3

**Summary:**

Summary:
This works falls into the line of work of using LLMs to generate programs to solve combinatorial optimization models. The authors incorporate additional constraints / priors in the generation process by converting the input CO problem into a bipartite graph as is commonly done. This graph is passed through a GNN to generate a “structure embedding” which is also passed as input to the final LLM.

**Questions:**

1.	How much does the performance of STRCMP depend on the underlying LLM? It would be interesting to see how performance changes with changing LLMs. This would help give an understanding of how much the structural embedding from the GNN contributes to the final result.

2.	Perozzi et al. [1] dive deeper into generating tokens from graphs and merging a GNN and LLM to improve downstream task performance. I would recommend giving their paper a read to look for potential avenues of improvement.

[1] Perozzi, B., Fatemi, B., Zelle, D., Tsitsulin, A., Kazemi, M., Al-Rfou, R. and Halcrow, J., 2024. Let your graph do the talking: Encoding structured data for llms. arXiv preprint arXiv:2402.05862.

**Ethical Concerns:**

["NO or VERY MINOR ethics concerns only"]

**Final Justification:**

Raising my score per the rebuttal. There are still some concern as raised by all reviewers but nevertheless an interesting paper with promising results that might worth sharing with the community.

**Limitations:**

Yes

**Paper Formatting Concerns:**

It seems ok

**Quality:**

2

**Strengths And Weaknesses:**

Strengths:
1.	The work is well motivated. Adding additional constraints to the generation process to improve quality of generation seems like a good avenue to explore.
2.	Paper is well written and structured

Weaknesses:
1.	Results are not convincing enough and a bit confusing in places. Across all results – Figs. 4, 5b, 8, 9, 10, 11 and Tables 5 and 6, the ablated variants of the STRCMP often outperform STRCMP. While the authors acknowledge this issue, I feel the paper is incomplete without an explanation as to why, or a way of picking which ablated variant to use in which case, especially given that in a lot of cases, the differences are quite drastic.
2.	Along the same lines, results on MILP instances shown in Figure 10 are not very convincing with LLM4Solver performing identical to STRCMP in most cases.

Please also consider positioning this work (and comparing where adequate) with recent LLM-based Optimization Co-Pilots. These seem omitted in the current version. Naming only a few here and plz see references therein to better position the work
- OptiMUS: Optimization Modeling Using MIP Solvers and large language models, A. AhmadiTeshnizi et. al.
- Constraint modelling with LLMs using in-context learning, K Michailidis, CP 2024
- Ner4Opt: Named entity recognition for optimization, Kadioglu et. al. Constraints 2024
- Enhancing decision making through the integration of large language models and operations research optimization, Wasserkrug et. al. AAAI'25
- Holy Grail 2.0: From Natural Language to Constraint Models, D. Tsouros et. al.

---

> ### Author Rebuttal · Authors · 2025-07-31
>
> We thank the reviewer for the insightful and valuable comments. We respond to your comments as follows and sincerely hope that our rebuttal could properly address your concerns. If so, we would deeply appreciate it if you could raise your score ("3: Borderline reject"). If not, please let us know your further concerns, and we will continue actively responding to your comments and improving our submission.
>
> **Q1: Ablated variants and performance: the paper is incomplete without an explanation as to why, or a way of picking which ablated variant to use in which case.**
>
> **A1:** Thank you for raising this critical question. We acknowledge that our initial draft did not sufficiently explain the performance variations among the different ablated versions of STRCMP, which could be confusing. We agree that providing a way to select the appropriate variant is crucial. In addition to the complete results for the SAT domain already in the appendix (Tables 5 and 6), we have added a comprehensive table detailing the optimization performance of the ablation variants on the MILP domain below.
>
> **Table 1: The Optimization Performance Result w.r.t. Primal-Dual Integral between Different Ablation Variants over MILP Domain**
> | Dataset| STRCMP| STRCMP (SFT Only) | STRCMP (DPO Only) | STRCMP w/o GNN |
> |:-|:-|:-|:-|:-|
> |corlat|**6597\.61**|6866\.61|7409\.85|7650\.13|
> |knapsack|3\.24|3\.23|3\.21|3\.19|
> |loadbalancing|2204\.89|**2121\.20**|2202\.50|2228\.26|
> |mik|269\.68|**252\.24**|284\.89|292\.41|
> |mis|5\.46|6\.64|5\.45|5\.48|
> |setcover|34\.02|33\.95|34\.17|36\.06|
>
> *Note that for the Primal-Dual Integral, a lower value indicates better performance.*
>
> Based on the complete test results across both SAT and MILP domains, we can draw three clear conclusions:
>
> 1.  **The structural prior is beneficial**: The `STRCMP w/o GNN` variant consistently underperforms compared to the other versions. This empirically validates our central hypothesis that integrating a structural prior into the LLM enhances its ability to generate high-quality code snippets for combinatorial optimization problems.
> 2.  **SFT is generally sufficient**: For most cases, `STRCMP (SFT Only)` achieves the best or near-best performance. This suggests that SFT alone is a highly effective method for adapting the composite model, providing a strong and efficient baseline.
> 3.  **Full post-training excels on harder instances**: For more complex problem instances, such as the `corlat` dataset (which is notable for its structural difficulty), the full `STRCMP` model (SFT+DPO) discovers a superior algorithm. This indicates that the preference optimization phase is particularly valuable when tackling challenging instances where nuanced trade-offs in the solution space are more critical.
>
> **Q2: Clarification on Experimental Results on MILP domain.**
>
> **A2:** Thank you for raising this point. We recognize that the smoothed curves in the original Figure 10 might have suggested that STRCMP's performance was identical to LLM4Solver in several cases. This was a simplification for visual clarity. On easier datasets like `knapsack`, `mis`, and `setcover`, the problems are not challenging enough to create significant performance separation between the advanced methods.
>
> To provide a clearer picture, we have compiled tables showing the non-smoothed, iteration-by-iteration performance on the more challenging MILP datasets. We employ an early stopping strategy where the search terminates if no performance improvement is observed within three consecutive iterations. As the tables below illustrate, our proposed STRCMP framework consistently discovers better-performing code snippets at an earlier stage compared to LLM4Solver, demonstrating its superior search efficiency.
>
> **Table 2: The Performance v.s. Iteration Result w.r.t. Primal-Dual Integral between Different Methods over corlat Dataset**
> | Iterations | LLM4Solver  | STRCMP      | STRCMP (SFT Only) | STRCMP (DPO Only) | STRCMP w/o GNN |
> | :- | :-  | :- | :-  | :- | :-  |
> | 3 | 9939\.48 | 8750\.84 | 6880\.25 | 7159\.24 | 10325\.77|
> | 6 | 9860\.20 | 8889\.19 | 7235\.31| 6870\.31| 10435\.85|
> | 9 | 9673\.47 | 8836\.85  | 7364\.47| 6881\.48|10404\.27|
> | 12| 9900\.06 |  | | 6932\.57 ||
>
> **Table 3: The Performance v.s. Iteration Result w.r.t. Primal-Dual Integral between Different Methods over mik Dataset**
> | Iterations | LLM4Solver  | STRCMP      | STRCMP (SFT Only) | STRCMP (DPO Only) | STRCMP w/o GNN |
> |:-|:-|:-|:-|:-|:-|
> |3|535\.04|478\.08|394\.25|451\.48|492\.96|
> |9|499\.60|455\.18|421\.44|434\.09|494\.24|
> |15|470\.48|450\.72|408\.55|431\.42|481\.97|
> |21|466\.45|438\.51|372\.12|||
> |27|471\.22|448\.31|380\.74|||
>
> **Table 4: The Performance v.s. Iteration Result w.r.t. Primal-Dual Integral between Different Methods over loadbalancing Dataset**
> | Iterations |LLM4Solver| STRCMP|STRCMP (SFT Only) | STRCMP (DPO Only) | STRCMP w/o GNN |
> |:-|:-|:-|:-|:-|:-|
> |3|1942\.27|1881\.74|2125\.22|2193\.62|1910\.53|
> |9|1885\.31|1878\.52|2065\.61|2245\.97|1905\.49|
> |15|1867\.71|1859\.14|2051\.36|||
>
> **Q3: Positioning this work (and comparing where adequate) with recent LLM-based Optimization Co-Pilots.**
>
> **A3:** We thank the reviewer for the insightful suggestions regarding LLM-based Optimization Co-Pilots and Perozzi et al.’s work. Below is our concise positioning:
>
> 1. **Positioning vs. LLM Optimization Co-Pilots**: Our work targets a distinct stage of the optimization pipeline: Co-Pilots (e.g., OptiMUS, Holy Grail 2.0) automate translation from natural language to formal models (front-end).  Our method enhances solver performance for pre-existing models (back-end), using LLMs to generate heuristics, branching strategies, and solver parameters. Thus, we complement—rather than overlap with—this emerging paradigm.
>
> 2. **Differentiation from Perozzi et al. [1]**: While both integrate GNNs and LLMs, our framework (STRCMP) diverges fundamentally: 1) **Objective**: Perozzi et al. focus on graph QA; our work targets synthesis of executable optimization algorithms; 2) **Training**: End-to-end training (used in [1]) proved unstable for our code-generation task (*please kindly see Table 4 in Response to Reviewer `zr9g`*). STRCMP uses two-stage training (pretrained GNN + frozen embeddings) followed by solver-guided refinement and 3) **Execution loop**: STRCMP embeds LLM output in an evolutionary feedback loop using solver execution—essential for generating high-performance CO code but absent in [1].
>
> **Q4: How much does the performance of STRCMP depend on the underlying LLM?**
>
> **A4:** Thank you for raising this insightful question. Your point about isolating the contribution of the GNN versus the LLM is well-taken. To address this, we have conducted additional ablation studies during the rebuttal period to evaluate the impact of different LLM backbones on STRCMP's performance.
>
> Given the time and resource constraints, we focused our evaluation on the Llama2 and Qwen2 families of models, selecting representatives of varying sizes. We performed training and testing on two datasets from the MILP domain and two from the SAT domain. The results are presented below.
>
> **Table 5: The Performance of STRCMP w.r.t. Primal-Dual Integral with Varied-Size Backbone LLM over MILP Domain**
> ||||STRCMP||STRCMP w/o GNN||
> |:-|:-|:-|:-|:-|:-|:-|
> |**Dataset**|Qwen2(0\.5B)|Qwen2(1\.5B)|Qwen2(7B)|Qwen2(14B)|Qwen2(7B)|Qwen2(14B)|
> |mik|--|--|269\.68|233\.56|292\.41|262\.74|
> |setcover|--|--|34\.02|34\.29|34\.06|34\.24|
>
> **Table 6: The Performance of STRCMP w.r.t. PAR-2 Score with Varied-Size Backbone LLM over SAT Domain**
> ||||STRCMP||STRCMP w/o GNN||
> |:-|:-|:-|:-|:-|:-|:-|
> |**Dataset**|Qwen2(0\.5B)|Qwen2(1\.5B)|Qwen2(7B)|Qwen2(14B)|Qwen2(7B)|Qwen2(14B)|
> |PRP|--|--|470\.63|414\.12|500\.00|455\.09|
> |Zamkeller|--|--|355\.3|127\.7|635\.75|308\.30|
>
> *Note that for both metrics, a lower value indicates better performance.*
>
> It is important to note that the entire Llama2 family of models failed on our code generation task, either by exceeding the context limitation of 4096 tokens or by being unable to generate syntactically correct, executable code. Similarly, the smaller Qwen2 models (0.5B and 1.5B) were also unable to produce viable code for the solvers. **This underscores the complexity of the task**, which requires a highly capable code-generating LLM. From the results in the tables, we can draw the following conclusions:
>
> 1.  **The structural embedding from the GNN consistently and significantly contributes to the final performance.** Across both MILP and SAT domains, and for both the 7B and 14B model sizes, STRCMP consistently outperforms its `STRCMP w/o GNN` variant. This confirms that the GNN provides a crucial inductive bias that guides the LLM toward better solutions, regardless of the LLM's scale.
>
> 2.  **STRCMP's performance is robust, provided a sufficiently capable LLM backbone is used.** While our framework's effectiveness is contingent on the LLM's fundamental ability to comprehend the task and generate valid code, the performance is stable across capable models of different sizes.
>
> 3.  **STRCMP's performance is sensitive to the size of the LLM backbone, but this effect is problem-dependent.** The results reveal a nuanced relationship between model scale and performance. For the more complex `mik` instances, scaling the LLM from 7B to 14B parameters yields a notable performance gain. However, for the `setcover` instances, the 7B and 14B models perform almost identically. This suggests that while larger models can unlock better performance on certain complex problem structures, a more moderately-sized model may be sufficient for others.
>
> We thank the reviewer for this suggestion, as these experiments have strengthened the paper and provided a clearer understanding of the interplay between the structural and language components of our framework. We will add this analysis to the final version of the paper.

---

> ### Author Response · Authors · 2025-08-05
>
> Dear Reviewer,
>
> We hope this message finds you well. As the discussion period concludes in less than two days, we wanted to confirm whether we have adequately addressed all your concerns.
>
> We have already incorporated feedback from Reviewers `2EbJ`, `7osJ`, and `zr9g`, who acknowledged our rebuttal and raised their scores accordingly. If you have any remaining questions or additional suggestions, we would greatly appreciate the opportunity to address them before the deadline.
>
> Your insights have been invaluable in improving our work, and we remain committed to refining it further based on your feedback.
>
> Thank you again for your time and thoughtful review.
>
> Best regards,
> The Authors of Submission7504

---

> > ### Comment · Reviewer_7TYC · 2025-08-05
> >
> > Thank you for your clarifications and additional experiments.

---

> > > ### Author Response · Authors · 2025-08-05
> > >
> > > Thank you for your response and for acknowledging our additional experiments. However, **we are still uncertain whether we have fully addressed all of your concerns**. If we have, we would greatly appreciate it if you could consider raising your score. If not, please do not hesitate to share any remaining concerns, and we will continue to actively engage with your feedback to further improve our submission.

---

### Official Review · Reviewer_zr9g · 2025-07-02

**Clarity:** 3
**Significance:** 3
**Originality:** 2
**Rating:** 4
**Confidence:** 5

**Summary:**

The paper provides a framework for solving combinatorial problems that integrates structure priors on the problems themselves to enhance solution quality and solving efficiency. This is achieved through combining a GNN to extract the structural embeddings from CO instances, which are then fed to a LLM that produces the solver-specific code.

**Questions:**

- This quote from Yang's analysis (lines 69-72) indeed points to a fundamental limitation of LLMs pre-trained primarily on text for structured domains. While fine-tuning is crucial, the initial pre-training distribution shapes the model's inherent priors. The STRCMP framework effectively addresses this by introducing structural priors via a GNN. Could you elaborate on how your "multi-modal co-learning" (line 223) specifically helps the LLM overcome these "weaker priors for structured domains"? Is it primarily the GNN providing the structured information that the LLM then learns to condition on, or is there a deeper, emergent understanding of structure fostered within the LLM itself due to this multi-modal input that goes beyond just adapting to the fine-tuning data?
- The paper mentions that the full STRCMP model (with SFT + DPO) doesn't uniformly outperform its ablations, suggesting "underlying conflicts within the post-training data distribution". Could you elaborate on what these "domain-specific distributional discrepancies" might entail?
- The paper has adopted a two-stage training due to "significant optimization challenges" with end-to-end joint training. Could you provide more specific details on these challenges? For instance, were they related to gradient propagation issues across the GNN-LLM boundary, computational resource limitations, or difficulties in aligning the learning objectives of the two very different modalities?
- The paper claims superior interpretability due to generating solver-executable code. However, CO problems can be highly complex, and the generated code snippets (e.g., hybrid_diversity_selector in Appendix I) can be quite intricate. How does the interpretability of these generated code functions scale with the increasing complexity of the CO problem or the sophistication of the heuristics discovered? Is the generated code always easily understandable and modifiable by a human domain expert, or can it become opaque in its own right, albeit in a "code" format?

**Ethical Concerns:**

["NO or VERY MINOR ethics concerns only"]

**Final Justification:**

The rebuttal has clarified some of the concerns.

**Limitations:**

The limitation of two stage process has been mentioned in the limitation. But this is also a major weakness of this paper.

**Quality:**

2

**Strengths And Weaknesses:**

**Strengths:**

- STRCMP consistently matches or exceeds all baseline performance in terms of solution optimality and computational efficiency. It shows significant reductions in timeouts and solving time for SAT problems, and maintaining strong performance parity for MILP
- The paper highlights that existing LLM-based approaches for CO problems "often neglect critical structural priors". STRCMP directly tackles this significant gap by explicitly integrating these structural priors.
- The paper also provides a theoretical analysis based on information theory and multi-modal co-learning, showing how integrating priors does not lower the upper bound on model performance.

**Weaknesses:**

- The ablations results are counterintuitive. STRCMP does not uniformly outperform its ablations (SFT Only and DPO Only) across all benchmarks. While acknowledged as a limitation and an area for future research, it indicates an incomplete understanding or unresolved issue in the current training protocol, which slightly diminishes the overall quality of the "full" proposed solution.
- The paper adopts a two-stage training procedure (GNN first, then a frozen GNN with LLM) due to "significant optimization challenges" with end-to-end joint training. This suggests a practical workaround rather than an ideal integrated solution. It might potentially limit the synergistic learning that could occur with true end-to-end training.
- The interpretability part is not convincing. While code generation offers interpretability over black-box NCO methods, the complexity of generated code for highly intricate CO problems or novel heuristics might still pose interpretability challenges in practice. The generated code, even if syntactically correct, might not always be easily digestible or modifiable by human experts without significant effort.

---

> ### Author Rebuttal · Authors · 2025-07-31
>
> We thank the reviewer for the insightful and valuable comments. We respond to your comments as follows and sincerely hope that our rebuttal could properly address your concerns. If so, we would deeply appreciate it if you could raise your score ("3: Borderline reject"). If not, please let us know your further concerns, and we will continue actively responding to your comments and improving our submission.
>
> **Q1: Ragarding the counterintuitive result of the ablation studies: could you elaborate on how your "multi-modal co-learning" specifically helps the LLM overcome these "weaker priors for structured domains"?**
>
> **A1:** We sincerely thank the reviewer for this insightful question, which highlights critical points regarding the significance of our proposed method’s improvements and the source of performance gains. To address your concerns, we have supplemented our experimental results from two perspectives.
>
> **(1) Comprehensive analysis of ablation variants across the MILP domain**, complementing the existing results on the SAT domain in Tables 5 and 6 of the appendix. This additional data helps disentangle the contributions of structural priors, SFT, and DPO to the performance of STRCMP, as shown below.
>
> **Table 1: The Optimization Performance Result w.r.t. Primal-Dual Integral between Different Ablation Variants over MILP Domain**
> | Dataset| STRCMP| STRCMP (SFT Only) | STRCMP (DPO Only) | STRCMP w/o GNN |
> |:-|:-|:-|:-|:-|
> |corlat|**6597\.61**|6866\.61|7709\.85|7650\.13|
> |knapsack|3\.24|3\.23|3\.21|3\.19|
> |loadbalancing|2204\.89|**2121\.20**|2202\.50|2228\.26|
> |mik|269\.68|**252\.24**|284\.89|292\.41|
> |mis|5\.46|6\.64|5\.45|5\.48|
> |setcover|34\.02|33\.95|34\.17|34\.06|
>
> *Note that for the Primal-Dual Integral, a lower value indicates better performance.*
>
> Based on the complete test results across both SAT and MILP domains, we can draw three clear conclusions:
>
> - **The structural prior (integrated via GNN) is critical to performance**: The `STRCMP w/o GNN` variant consistently underperforms compared to the other versions. This empirically validates our central hypothesis that integrating a structural prior into the LLM enhances its ability to generate high-quality code snippets for combinatorial optimization problems.
> - **SFT is generally sufficient**: For most cases, `STRCMP (SFT Only)` achieves the best or near-best performance. This suggests that SFT alone is a highly effective method for adapting the composite model, providing a strong and efficient baseline.
> - **Full post-training excels on harder instances**: For more complex problem instances, such as the `corlat` dataset (which is notable for its structural difficulty), the full `STRCMP` model (SFT+DPO) discovers a superior algorithm. This indicates that the preference optimization phase is particularly valuable when tackling challenging instances where nuanced trade-offs in the solution space are more critical.
>
> **(2) Experiments about isolating the contribution of the GNN versus the LLM backbone.** To address this, we have conducted additional ablation studies during the rebuttal period to evaluate the impact of different LLM backbones on STRCMP's performance. Given the time and resource constraints, we focused our evaluation on the Llama2 and Qwen2 families of models, selecting representatives of varying sizes. We performed training and testing on two datasets from the MILP domain and two from the SAT domain. The results are presented below.
>
> **Table 2: The Performance of STRCMP w.r.t. Primal-Dual Integral with Varied-Size Backbone LLM over MILP Domain**
> ||||STRCMP||STRCMP w/o GNN||
> |:-|:-|:-|:-|:-|:-|:-|
> |**Dataset**|Qwen2(0\.5B)|Qwen2(1\.5B)|Qwen2(7B)|Qwen2(14B)|Qwen2(7B)|Qwen2(14B)|
> |mik|--|--|269\.68|233\.56|292\.41|262\.74|
> |setcover|--|--|34\.02|34\.29|34\.06|34\.24|
>
> **Table 3: The Performance of STRCMP w.r.t. PAR-2 Score with Varied-Size Backbone LLM over SAT Domain**
> ||||STRCMP||STRCMP w/o GNN||
> |:-|:-|:-|:-|:-|:-|:-|
> |**Dataset**|Qwen2(0\.5B)|Qwen2(1\.5B)|Qwen2(7B)|Qwen2(14B)|Qwen2(7B)|Qwen2(14B)|
> |PRP|--|--|470\.63|414\.12|500\.00|455\.09|
> |Zamkeller|--|--|355\.3|127\.7|635\.75|308\.30|
>
> *Note that for both metrics, a lower value indicates better performance.*
>
> It is important to note that the entire Llama2 family of models failed on our code generation task, either by exceeding the context limitation of 4096 tokens or by being unable to generate syntactically correct, executable code. Similarly, the smaller Qwen2 models (0.5B and 1.5B) were also unable to produce viable code for the solvers. This underscores the complexity of the task, which requires a highly capable code-generating LLM. From the results in the tables, we can draw the following conclusions: **the structural embedding from the GNN consistently and significantly contributes to the final performance.** Across both MILP and SAT domains, and for both the 7B and 14B model sizes, STRCMP consistently outperforms its `STRCMP w/o GNN` variant. This confirms that the GNN provides a crucial inductive bias that guides the LLM toward better solutions, regardless of the LLM's scale.
>
> **Q2: Could you provide more specific details on challenges related to joint end-to-end training?**
>
> **A2:** We sincerely thank the reviewer for this insightful question. We admit that the initial submission lacked a thorough discussion of why we adopted the two-stage training procedure. To fully and convincingly answer this question, we have re-designed and implemented an end-to-end joint training method for our proposed model. This new method directly aligns the learning objectives of the graph and text modalities with the same optimization goal: improving the performance of the generated code. Specifically, we do not discriminate between the weights of the GNN and the LLM, meaning they share the same loss function, and the gradient is backpropagated from the parameters of the LLM to those of the GNN.
>
> We implemented the end-to-end joint training method described above and tested the resulting models on two SAT datasets, given the time constraints of the rebuttal phase. *All the experimental settings keep the same except for the training method*. The results of this new training and testing are presented below. We can conclude two main points from these results:
>
> 1.  **The end-to-end joint training is extremely unstable compared to our proposed two-stage training method.** We infer that the training instability stems from the architectural and optimization differences between the GNN and the LLM. The gradients, originating from a sequence-level generation loss in the deep LLM, become noisy and attenuated when backpropagated through the entire LLM and into the much shallower GNN. This creates a challenging optimization landscape where a single learning rate is ineffective for both components, leading to unstable convergence (even with many more training steps) as shown in Table 4. Compared with the joint training, **the loss curve of our two-stage training method is stable and close to zero at convergence** (Please see the loss curve of our two-stage training method in Figure 7 in Appendix B). The GNN and LLM are trained together, which makes it difficult for the models to learn optimal parameters simultaneously.
>
> **Table 4: The Training Loss of End-to-End Joint Training**
> |**Training Step**|**End-to-End DPO Loss**|**End-to-End SFT Loss**|
> |:-|:-|:-|
> |100|0\.54|1\.64|
> |200|0\.60|1\.53|
> |300|0\.26|1\.44|
> |400|0\.43|1\.28|
> |500|0\.56|1\.14|
> |600|0\.13|1\.04|
> |700|0\.42|0\.98|
> |800|0\.29|0\.88|
> |900|0\.30|0\.86|
>
> 2.  **The model exhibits poor performance when tested on solving SAT instances.** Unsurprisingly, due to its very unstable training and relatively high loss at convergence, the performance of the jointly trained model is lower than our two-stage model on the two SAT datasets, as shown in Table 5.
>
> **Table 5: The Jointly Trained Model's Performance over Two SAT datasets** w.r.t. PAR-2 Score
> | **Dataset** | **STRCMP (Two-Stage)** | **Joint End-to-End Model** |
> | :--- | :--- | :--- |
> | **Zamkeller** | 270.62 | 751.34 |
> | **PRP** | 1804.46 | 2092.71 |
>
> *Note that for PAR-2 Score, a lower value indicates better performance.*
>
> These empirical findings confirm the "significant optimization challenges" mentioned in the paper and **justify our decision to use a more stable and effective two-stage training protocol**.
>
> **Q3: How does the interpretability of these generated code functions scale with the increasing complexity of the CO problem or the sophistication of the heuristics discovered?**
>
> **A3:** We appreciate your insightful inquiry regarding the scalability of interpretability in STRCMP. We fully acknowledge that while code generation enhances transparency relative to neural black-box methods, interpretability does not scale linearly with problem complexity. For highly intricate CO problems or novel heuristics, generated code may involve nested logic, context-dependent optimizations, or non-intuitive transformations. This necessitates non-trivial effort for human experts to parse, reducing immediate interpretability.
>
> Nevertheless, STRCMP retains fundamental advantages over neural NCO methods:  1) **Inspectability**: solver logic remains open to direct inspection in its artifactual form; 2) **Modifiability**: code can be edited without model retraining; and 3) **Debuggability**: runtime validation and issue tracing via standard tools remain possible.
>
> We view the interpretability of complex generated code as an ongoing research challenge and are committed to addressing it in future work. Potential directions include simplifying the generated algorithms through post-processing or enhancing them with annotations to aid human understanding. For now, we believe STRCMP strikes a valuable balance, advancing interpretability over NCO methods while laying the groundwork for further improvements.

---

> > ### Comment · Reviewer_zr9g · 2025-08-03
> > **Questions about rebuttal**
> >
> > Thanks for the rebuttal and the new experiments. Please note the followings:
> > - Q1 has not been discussed in the rebuttal (the rebuttal starts from Q2).
> > - The new experiments of the ablation study still raises this question of whether we need a model with  SFT+DPO. It works only for one dataset. Also, what is the complexity of the dataset? Could you provide insights on when one might need SFT+DPO?
> > - Interpretability:  My understanding is that STRCMP has some advantages on other methods but necessarily on interpretability.

---

> ### Author Response · Authors · 2025-08-04
> **Response to Reviewer's Comment on the Effectiveness of Structural Prior (Q1)**
>
> Thank you for your valuable feedback and for allowing us to clarify our contributions. We apologize for the previous ambiguity in our rebuttal. We provide a distinct and detailed discussion for Q1 below, supported by new experimental results. We sincerely appreciate you raising this insightful point. Here, we address your question: **Could you elaborate on how your "multi-modal co-learning" (line 223) specifically helps the LLM overcome these "weaker priors for structured domains"? Is it primarily the GNN providing the structured information that the LLM then learns to condition on?**
>
> ---
>
> To empirically validate the criticality of this GNN-provided structural prior, we have re-run our ablation experiments with a more robust termination condition.
>
> ## Additional Experimental Validation over Two SAT Datasets.
>
> We re-ran the experiments for the CNP and CoinsGrid datasets on the SAT domain since in the original experiment, the performance distinction between different variants of STRCMP is very close, which cannot correctly distinguish the performance of different variants especially the effectiveness of inclusion of GNN. The key change was switching the termination condition from a fixed maximum number of iterations to stopping when the model's training set performance did not improve for three consecutive iterations. This aligns the methodology with our MILP experiments and provides a clearer view of each variant's potential. The new results are presented in Table 1.
>
> **Table 1: The Optimization Performance result w.r.t. PAR-2 Score between Different Ablation Variants over SAT Domain**
>
> | **Dataset** | **STRCMP** | **STRCMP (SFT Only)** | **STRCMP (DPO Only)** | **STRCMP w/o GNN** |
> |:---|:---:|:---:|:---:|:---:|
> |CNP|27274.83|26677.00|18752.52|29368.38|
> |CoinsGrid|45315.27|47635.57|31444.33|48148.67|
>
> ## Experimental Validation over MILP Datasets
>
> **Table 2: The Optimization Performance result w.r.t. Primal-Dual Integral between Different Ablation Variants over MILP domain**
> | Dataset| STRCMP| STRCMP (SFT Only) | STRCMP (DPO Only) | STRCMP w/o GNN |
> |:-|:-|:-|:-|:-|
> |corlat|**6597\.61**|6866\.61|7709\.85|7650\.13|
> |knapsack|3\.24|3\.23|3\.21|3\.19|
> |loadbalancing|2204\.89|**2121\.20**|2202\.50|2228\.26|
> |mik|269\.68|**252\.24**|284\.89|292\.41|
> |mis|5\.46|6\.64|5\.45|5\.48|
> |setcover|34\.02|33\.95|34\.17|34\.06|
>
> ## Experimental Validation on Structral Prior's Contribution with Varied-Size backbone LLM
>
> **Table 3: The Performance Comparison between STRCMP and STRCMP w/o GNN w.r.t. Primal-Dual Integral with Varied-Size Backbone LLM over MILP Domain**
> ||STRCMP||STRCMP w/o GNN||
> |:-|:-|:-|:-|:-|
> |**Dataset**|Qwen2(7B)|Qwen2(14B)|Qwen2(7B)|Qwen2(14B)|Qwen2(7B)|Qwen2(14B)|
> |mik|269\.68|233\.56|292\.41|262\.74|
> |setcover|34\.02|34\.29|34\.06|34\.24|
>
> **Table 4: The Performance Comparison between STRCMP and STRCMP w/o GNN w.r.t. PAR-2 score with Varied-Size Backbone LLM over SAT Domain**
> ||STRCMP||STRCMP w/o GNN||
> |:-|:-|:-|:-|:-|
> |**Dataset**|Qwen2(7B)|Qwen2(14B)|Qwen2(7B)|Qwen2(14B)|Qwen2(7B)|Qwen2(14B)|
> |PRP|470\.63|414\.12|500\.00|455\.09|
> |Zamkeller|355\.3|127\.7|635\.75|308\.30|
>
> ---
>
> ## Claims Made Upon above Results
> By combining these updated results with our existing experiments (Tables 2, 3, and 4), we can draw two firm conclusions:
>
> 1.  **The structural prior (integrated via GNN) is critical to performance**. In every experiment across both SAT (Table 1) and MILP (Table 2) domains, the `STRCMP w/o GNN` variant consistently underperforms compared to the full STRCMP model and its other variants with GNN. This empirically validates our central hypothesis that integrating a structural prior is essential for enhancing the LLM's ability to solve combinatorial optimization problems.
>
> 2.  **The structural embedding from the GNN consistently and significantly contributes to the final performance.** As shown in Tables 3 and 4, STRCMP consistently outperforms its `STRCMP w/o GNN` counterpart across both the 7B and 14B model sizes. This demonstrates that the GNN provides a crucial and scalable inductive bias that guides the LLM toward better solutions, regardless of the LLM's parameter count.
>
> These comprehensive results strongly support our claim that the multi-modal co-learning approach, driven by the GNN's structural prior, is the key factor in overcoming the LLM's inherent weaknesses in structured domains.

---

> ### Author Response · Authors · 2025-08-04
> **Response to Reviewer's Comment on the Training Strategy (Q2) (1/2)**
>
> **Q2: The new experiments of the ablation study still raises this question of whether we need a model with SFT+DPO. It works only for one dataset. Also, what is the complexity of the dataset? Could you provide insights on when one might need SFT+DPO?**
>
> **A2:** To provide a clearer and more comprehensive answer, we offer a detailed analysis focusing on dataset complexity and the iterative optimization behavior of our proposed method, STRCMP, and its variants.
>
> ## Dataset Complexity
> The complexity of combinatorial optimization problems can be characterized by various metrics. Drawing from established literature, we focus on several key graph-based metrics—Modularity, Density, and Clustering—to provide a quantitative perspective on the structural complexity of the datasets used in our experiments. *While not absolute, higher values in these metrics generally suggest a more intricate problem structure and complexity*. The problem sizes are detailed in Appendix D of our manuscript. The statistics for the MILP and SAT datasets are presented below.
>
> **Table 1: Data Complexity of MILP Dataset**
> |**Dataset**|**Modularity**|**Density**|**Clustering**|
> |:-|:-|:-|:-|
> |setcover|0.18|0.07|0.34|
> |knapsack|0.07|0.04|0.36|
> |mis|0.48|0.01|0.02|
> |mik|0.11|0.23|0.30|
> |loadbalancing|0.49|0.01|0.01|
>
> **Table 2: Data Complexity of SAT Dataset**
> |**Dataset**|**Modularity**|**Density**|**Clustering**|
> |:-|:-|:-|:-|
> |CNP|0\.8181|0\.0016|0\.2012|
> |CoinsGrid|0\.9306|0\.0005|0\.5377|
> |Zamkeller|0\.7383|0\.0004|0\.5659|
> |PRP|0.8865|0.0002|0.5429|
> ## Iterative Behavior of STRCMP and its Variants
> In addition to the iterative optimization results for the MILP domain (presented in Tables 2, 3, and 4 in our main rebuttal to Reviewer `7TYC`), we now provide the corresponding analysis for the SAT domain. These results illustrate the performance trajectory (PAR-2 score) of STRCMP against its SFT-only and DPO-only ablations.
>
> *Note: We employ a conditional stopping strategy, halting the training process if the model's performance on the training set does not improve for three consecutive iterations. Hence the total iteration number of different method will be varied.*
>
> **Table 3: The Performance v.s. Iteration Result w.r.t. PAR-2 Score between Different Methods over CNP Dataset**
> |**Iteration**|**STRCMP**|**STRCMP (SFT Only)**|**STRCMP (DPO Only)**|
> |:-|:-|:-|:-|
> |3|49931|49395\.1667|48963\.5|
> |15|60433\.3333|59101\.3333|57184|
> |…|…|…|…|
> |51|57108\.1667|51935\.6667|57688\.67|
> |57|55257\.8333|52890\.1667|57352|
> |…|…|…|…|
> |75|||48779\.5|
> |78|||47443|
>
> **Table 4: The Performance v.s. Iteration Result w.r.t. PAR-2 Score between Different Methods over CoinsGrid Dataset**
> |**Iteration**|**STRCMP**|**STRCMP (SFT Only)**|**STRCMP (DPO Only)**|
> |:-|:-|:-|:-|
> |3|60376|59444|59735\.67|
> |9|65177\.67|65305|70214\.83|
> |…|…|…|…|
> |30|58516\.83|62026\.17|61296\.83|
> |33|58091\.67|62460\.83|61622\.17|
> |…|…|…|…|
> |57||56775\.17|53737\.67|
> |60|||58212\.83|
>
> **Table 5: The Performance v.s. Iteration Result w.r.t. PAR-2 Score between Different Methods over PRP Dataset**
> |**Iteration**|**STRCMP**|**STRCMP (SFT Only)**|**STRCMP (DPO Only)**|
> |:-|:-|:-|:-|
> |3|53808\.17|53696\.33|54977\.5|
> |9|54523|56456\.67|54462\.5|
> |…|…|…|…|
> |39|53015\.83|55141\.67|52635|
> |42|50812\.5|53459|52731\.67|
> |…|…|…|…|
> |66|52701\.5|53627|51030|
> |69|||50352\.33|
> |…|…|…|…|
> |87|||49281|
>
> **Table 6: The Performance v.s. Iteration Result w.r.t. PAR-2 Score between Different Methods over Zamkeller Dataset**
> |**Iteration**|**STRCMP**|**STRCMP (SFT Only)**|**STRCMP (DPO Only)**|
> |:-|:-|:-|:-|
> |3|45613\.5|44988|45427|
> |6|56487\.17|47457\.5|51138|
> |…|…|…|…|
> |27|56511\.5|49499\.17|44826\.5|
> |30|54933\.67|50914|45185\.67|
> |…|…|…|…|
> |51|51801\.33|53002\.83|42444\.17|
> |54|51900\.67|49976\.5|38346\.17|
> |…|…|…|…|
> |72|||41710|
> |75|||36365|

---

> ### Author Response · Authors · 2025-08-04
> **Response to Reviewer's Comment on the Training Strategy (Q2) (2/2)**
>
> ## Analysis and Insights
> From the iterative performance results, we draw two primary observations:
>
> 1. **SFT-only models often converge faster in terms of iteration count but may stabilize at a suboptimal performance level**. This suggests that while SFT is effective at rapidly imitating strong solutions, it may overfit to the strategies present in the initial dataset, leading to premature convergence in a local optimum.
>
> 2. **DPO-inclusive models (DPO-only and SFT+DPO) demonstrate a capacity for continued improvement over a longer training horizon**, often surpassing the SFT-only model's peak performance. We posit that DPO enhances the model's generalization capabilities. By learning from preference pairs rather than absolute solutions, the model develops a more nuanced understanding of what constitutes a better search trajectory, allowing it to escape local optima and discover superior solutions.
>
> Based on these observations and the complexity analysis, we can **summarize our insights on selecting a training strategy**:
>
> 1.  **For "easy" datasets** (e.g., small-scale, sparse, solvable to optimality quickly): The **SFT**-only approach is often sufficient and efficient. On these problems, high-quality training data (i.e., optimal or near-optimal solutions) can be generated at a low cost, allowing **SFT** to quickly learn an effective policy.
> 2.  **For "hard" datasets** (e.g., large-scale, dense, complex structure, intractable): We strongly recommend a **DPO**-based approach (**DPO**-only or **SFT+DPO**). For these problems, obtaining optimal solutions for **SFT** is prohibitively expensive or impossible. However, generating preference pairs by comparing the performance of different candidate solutions is still feasible and relatively cheap. The superior generalization of **DPO**-trained models makes them better suited to navigating the vast and complex search spaces of these challenging instances.
>
> 3. **For medium complexity datasets, the choice is less definitive**. As the results show, the interplay between SFT and DPO is intricate. Our full STRCMP model (SFT+DPO) often acts as a robust default, leveraging SFT for a strong initialization and DPO for refinement and generalization. However, we acknowledge that determining the optimal blend is a significant challenge. **The task-specific training of large models has not yet fully clarified the precise relationship between SFT and DPO—when one is definitively superior, or how their data should be optimally combined.** We consider this an important open problem for the field.
>
> We hope this detailed analysis clarifies the rationale behind our approach and provides valuable insights into when a combined SFT+DPO model is most impactful.
>
> ---
>
> Reference
>
> [1] Li, Yang, et al. "Hardsatgen: Understanding the difficulty of hard sat formula generation and a strong structure-hardness-aware baseline." Proceedings of the 29th ACM SIGKDD Conference on Knowledge Discovery and Data Mining. 2023.
>
> [2] Geng, Zijie, et al. "A deep instance generative framework for milp solvers under limited data availability." Advances in Neural Information Processing Systems 36 (2023): 26025-26047.
>
> [3] Ye, Huigen, et al. "ML4MILP: A Benchmark Dataset for Machine Learning-based Mixed-Integer Linear Programming."

---

> ### Author Response · Authors · 2025-08-04
> **Response to Reviewer's Comment on Interpretability​ (Q3)**
>
> While our proposed method, STRCMP, does exhibit certain advantages in interpretability when compared to some existing neural combinatorial methods, this was not the primary focus of our work. We acknowledge that a comprehensive and rigorous treatment of the interpretability of deep learning models for combinatorial optimization is a significant and challenging research area. Fully addressing this issue would require a dedicated study that, we believe, extends beyond the intended scope of the current paper. We appreciate the reviewer highlighting this important aspect, which we consider a valuable direction for future research.

---

> ### Author Response · Authors · 2025-08-04
> **General Response to Questions about Initial Rebuttal**
>
> We sincerely hope this detailed response has adequately addressed your remaining concerns. If so, we would deeply appreciate it if you could raise your score ("3: Borderline reject"). If not, please let us know your further concerns, and we will continue actively responding to your comments and improving our submission. We are grateful for the thorough and constructive feedback throughout this process, as it has helped us significantly strengthen the paper.

---

> > ### Comment · Reviewer_zr9g · 2025-08-05
> >
> > Thank you for the rebuttal. I have adjusted my score accordingly. Please include these responses in the revised version.

---

> ### Author Response · Authors · 2025-08-05
> **Many Thanks for the Positive Discussion and Constructive Suggestions**
>
> Thank you for your response and for acknowledging our work. We greatly appreciate the time and effort you have devoted to reviewing it. If you have any further questions or concerns, please feel free to let us know—we would be happy to address them. We will incorporate all feedback into the final version of our paper.

---

### Author Response · Authors · 2025-08-08
**General Response (1/2)**

We would like to extend our sincere gratitude for your valuable feedback and constructive suggestions. For your convenience, we have prepared a summary of our responses and outlined how we have addressed the reviewers' concerns as follows. We sincerely hope that this summary will facilitate your review and lighten your workload.

Our paper has received many encouraging **positive feedbacks** from the reviewers, such as "**STRCMP directly tackles this significant gap/strong performance**" (Reviewer `zr9g`), "**well motivated**" (Reviewers `7TYC` and `2Ebj`), "**a good avenue to explore**" (Reviewer `7TYC`), "**a promising general idea**" and "**good writing and sufficient details**" (Reviewers `7osj` and `2Ebj`). "**I believe this work serves as a bridge connecting GNN and LLM in CO problems, which is novel and has its value**" (Reviewer `2Ebj`).

We outline how we have addressed the concerns raised by each reviewer as follows.

## Common Concerns

> **(Reviewers `zr9g` and `2Ebj`) 1. The reason why adopted a two-phase training strategy instead of joint end-to-end training.**

We sincerely thank the reviewer for this insightful question. We admit that the initial submission lacked a thorough discussion of why we adopted the two-stage training procedure. To fully and convincingly answer this question, we have **re-designed and implemented an end-to-end joint training method for our proposed model**.

We implemented the end-to-end joint training method described above and tested the resulting models on two SAT datasets, given the time constraints of the rebuttal phase. All the experimental settings keep the same except for the training method. We can conclude two main points from this experiment:

1. **The end-to-end joint training is extremely unstable compared to our proposed two-stage training method**. We infer that the training instability stems from the architectural and optimization differences between the GNN and the LLM. The gradients, originating from a sequence-level generation loss in the deep LLM, become noisy and attenuated when backpropagated through the entire LLM and into the much shallower GNN. Compared with the joint training, the loss curve of our two-stage training method is stable and close to zero at convergence. The GNN and LLM are trained together, which makes it difficult for the models to learn optimal parameters simultaneously.

2. **The model exhibits poor performance when tested on solving SAT instances**. Unsurprisingly, due to its very unstable training and relatively high loss at convergence, the performance of the jointly trained model is lower than our two-stage model on the two SAT datasets.

> **(Reviewers `zr9g` and `7TYC`) 2. The insight of training strategy for the proposed STRCMP.**

To provide a clearer and more comprehensive answer, we offer a detailed analysis focusing on dataset complexity and the iterative optimization behavior of our proposed method, STRCMP, and its variants. Through the above thorough analysis and additional experiments, we can summarize our insights on selecting a training strategy:

1.  **For "easy" datasets** : The **SFT**-only approach is often sufficient and efficient. On these problems, high-quality training data (i.e., optimal or near-optimal solutions) can be generated at a low cost, allowing **SFT** to quickly learn an effective policy.

2.  **For "hard" datasets** : We strongly recommend a **DPO**-based approach (**DPO**-only or **SFT+DPO**). For these problems, obtaining optimal solutions for **SFT** is prohibitively expensive or impossible. However, generating preference pairs by comparing the performance of different candidate solutions is still feasible and relatively cheap. The superior generalization of **DPO**-trained models makes them better suited to navigating the vast and complex search spaces of these challenging instances.

3. **For medium complexity datasets, the choice is less definitive**. As the results show, the interplay between SFT and DPO is intricate. Our full STRCMP model (SFT+DPO) often acts as a robust default, leveraging SFT for a strong initialization and DPO for refinement and generalization.

> **(Reviewers `zr9g` and `7TYC`) 3. Please further explain the result of the ablation studies.**

To address this concern, we have supplemented our experimental results in terms of 1) comprehensive analysis of ablation variants across the MILP domai and 2) experiments about isolating the contribution of the GNN versus the LLM backbone. Above additional experiments demontrates that 1) **the structural prior (integrated via GNN) is critical to performance** and 2) **the structural embedding from the GNN consistently contributes to the final performance with the varied backbone LLMs**.

---

> ### Author Response · Authors · 2025-08-08
> **General Response (2/2)**
>
> ## Reviewer `zr9g`
>
> > **4. Interpretability of these generated code functions scale with the increasing complexity of the CO problem.**
>
> We explain the interpretability of the generated code functions with the following points: 1) **Inspectability**: solver logic remains open to direct inspection in its artifactual form; 2) **Modifiability**: code can be edited without model retraining; and 3) **Debuggability**: runtime validation and issue tracing via standard tools remain possible. We view the interpretability of complex generated code as an ongoing research challenge and are committed to addressing it in future work.
>
> ## Reviewer `7TYC`
> > **5. Clarification on Experimental Results on MILP domain.**
>
> To provide a clearer picture, we have compiled tables showing the non-smoothed, iteration-by-iteration performance on the more challenging MILP datasets. We employ an early stopping strategy where the search terminates if no performance improvement is observed within three consecutive iterations. As the newly added results shown, our proposed STRCMP framework **consistently discovers better-performing code snippets at an earlier stage compared to LLM4Solver**, demonstrating its superior search efficiency.
>
> > **6. Positioning this work (and comparing where adequate) with recent LLM-based Optimization Co-Pilots.**
>
> We have supplemented the thorough comparison between our work and reviewer listed work.
>
> > **7. How much does the performance of STRCMP depend on the underlying LLM?**
>
> To address this, we have conducted **additional ablation studies during the rebuttal period to evaluate the impact of different LLM backbones on STRCMP's performance**. Given the time and resource constraints, we focused our evaluation on the Llama2 and Qwen2 families of models, selecting representatives of varying sizes. We performed training and testing on two datasets from the MILP domain and two from the SAT domain.
>
> ## Reviewer `7osj`
>
> > **8. The advantage of STRCMP over existing framework (LLM4Solver and AutoSAT) in terms of efficiency and effectiveness.**
>
> We have highlighted the distinct improvements in both efficiency and effectiveness relative to existing algorithm discovery paradigms, **specifically achieving superior or equivalent solution quality with reduced computational resources**.
>
> > **9. Adjustment of problem setting.**
>
> We have tried to addjust the problem setting. Following this new setting, we have conducted a new set of experiments to validate this revised problem-specific framework. The experimental results demonstrate that this approach is highly effective and validate the reviewer's intuition and strengthens our paper's core thesis: **fusing structural information with LLM-based code generation is a powerful technique for discovering tailored, high-performance algorithms**.
>
> ## Reviewer `2Ebj`
>
> > **10. Please explain the natural language description of the CO problem.**
>
> We have explained and give more details about the natural language description of the CO problem.
>
> > **11. Justification on that different problems subsume different structures.**
>
> To further justify this methodology and directly respond to this concern, we conducted an experiment comparing two training strategies. We trained a separate STRCMP model using only one class of dataset and compared its performance with that of a model trained on a mixed dataset encompassing multiple CO problems. The experimental results highlight that training on a mixed-class dataset **not only mitigates the risk of overfitting to a single problem’s structure but also enhances generalization and efficiency across structurally similar CO problems**.
>
> > **12. The executability and correctness of LLM-generated code in the proposed framework.**
>
> We have explained the executability and correctness of LLM-generated code in terms of compilation check and performance evaluation.
>
> > **13. Explain more about the sup(P) in definition 1.**
>
>  We have explained this definition via breaking down the formula sup(P) and explaining the components seperately.

---

### Note · Authors · 2025-08-11

Dear Area Chair and Reviewers,

We sincerely thank you for the invaluable feedback and the constructive discussion period. We have prepared a summary of our responses and outlined how we have addressed the reviewers' concerns. Please see the last one official comment (i.e., **General Response**).

We are encouraged that our rebuttal has effectively addressed the reviewers' primary concerns, leading to a **positive reassessment** from the review team. For your convenience, the updated status of the reviews is summarized below:

| Reviewer                 | zr9g                                      | 7osJ                     | 2EbJ                                 | 7TYC                                                  |
| :----------------------- | :---------------------------------------- | :----------------------- | :----------------------------------- | :---------------------------------------------------- |
| Original Rating        | 3: Borderline reject                      | 3: Borderline reject     | 4: Borderline accept                 | 3: Borderline reject                               |
| Willing to Raise Score | **Yes**                                       | **Yes**                      | **Yes**                                  | TBD                                                   |
| Related Comments       | "**I have adjusted my score accordingly.**" | "**I have raised my score**" | "**I would like to raise my scores.**" | "Thank you for your clarifications and additional experiments." |

With respect to the final comments from Reviewer `7TYC`, *we are uncertain whether we have fully addressed all his concerns*. However, we still want to thank him for his valuable feedback and suggestions. We hope our clarifications were helpful, and we value the feedback, which has undoubtedly strengthened our work.

We trust that these clarifications and the reviewers' positive reception will facilitate your final decision-making process. Thank you for your time and consideration.

Best regards,

The Authors

---

### Decision · Program_Chairs · 2025-09-17

**Decision:**

Accept (poster)

**Comment:**

In this paper, the authors introduce STRCMP, a framework that integrates graph structural priors with LLMs to address combinatorial optimization problems. By embedding explicit graph information into LLM-driven reasoning, STRCMP improves the model’s ability to capture problem structure in tasks such as routing and scheduling. Experimental results show that STRCMP delivers competitive or superior performance compared to heuristic solvers and LLM-only methods, which demonstrates the effectiveness of combining graph priors with LLM adaptability. After rebuttal, all reviewers turn to be positive about this paper, and I also agree with them for acceptance. However, please take into account the comments of reviewers and the new results during rebuttal in the final version.